# Online Variational Filtering and Parameter Learning

**Andrew Campbell** *    **Yuyang Shi** *    **Tom Rainforth**    **Arnaud Doucet**
Department of Statistics, University of Oxford, UK
{campbell, yshi, rainforth, doucet}@stats.ox.ac.uk

## Abstract

We present a variational method for *online* state estimation and parameter learning in state-space models (SSMs), a ubiquitous class of latent variable models for sequential data. As per standard batch variational techniques, we use stochastic gradients to simultaneously optimize a lower bound on the log evidence with respect to both model parameters and a variational approximation of the states' posterior distribution. However, unlike existing approaches, our method is able to operate in an entirely online manner, such that historic observations do not require revisitation after being incorporated and the cost of updates at each time step remains constant, despite the growing dimensionality of the joint posterior distribution of the states. This is achieved by utilizing backward decompositions of this joint posterior distribution and of its variational approximation, combined with Bellman-type recursions for the evidence lower bound and its gradients. We demonstrate the performance of this methodology across several examples, including high-dimensional SSMs and sequential Variational Auto-Encoders.

## 1    Introduction

Many tasks in machine learning with time series data—such as video prediction [16, 27], speech enhancement [35] or robot localization [9, 18, 28]—often need to be performed online. Online techniques are also necessary in contexts as diverse as target tracking [4], weather prediction [13] and financial forecasting [42]. A popular class of models for these sequential data are SSMs which, when combined with neural network ideas, can also be used to define powerful sequential Variational Auto-Encoders (VAEs); see e.g. [8, 15, 16, 30]. However, performing inference in SSMs is a challenging problem and approximate inference techniques for such models remain an active research area.

Formally, an SSM is described by a latent Markov process and an observation process. Even if the model parameters are assumed known, online inference of the states of the latent process is a complex problem known as filtering. Standard approximations such as the extended Kalman Filter (KF), ensemble KF, and unscented KF can be used, but only provide an ad hoc Gaussian approximation to the filter [13, 17, 37]. More generally, assumed density filtering techniques [3, 7, 32] can provide other simple parametric family approximations. These approximate filtering methods can be used, in turn, to develop online parameter learning procedures by either augmenting the state with the static parameters or using gradient-based approaches. However, such approaches are notoriously unreliable [19]. Particle Filtering (PF) methods, on the other hand, provide a more principled approach for online state and parameter estimation with theoretical guarantees [11, 12, 19, 41], but the variance of PF estimates typically scales exponentially with the state dimension [6].

Although they typically do not return consistent estimates, variational techniques provide an attractive alternative for simultaneous state estimation and parameter learning which scales better to high-dimensional latent states than PF, and are not restricted to simple parametric approximations. Many

---

*Equal contribution

35th Conference on Neural Information Processing Systems (NeurIPS 2021).

such methods have been proposed for SSMs over recent years, e.g. [2, 10, 20, 22, 23, 29, 33, 36]. However, they have generally been developed for *batch inference* where one maximizes the Evidence Lower Bound (ELBO) for a *fixed* dataset. As such, they are ill-suited for *online* learning as, whenever a new observation is collected, one would need to update the entire joint variational states distribution whose dimension increases over time. Though a small number of online variational approaches have been developed [30, 38, 45], these rely on significant restrictions of the variational family, leading to approximations that cannot faithfully approximate the posterior distribution of the latent states.

The main contribution of this paper is a novel variational approach to perform *online* filtering and parameter learning for SSMs which bypasses those restrictions. As per standard batch variational inference, we simultaneously maximize an ELBO with respect to both model parameters and a variational approximation of the joint state posterior. However, our method operates in an entirely online manner and the cost of updates at each time step remains constant. Key to our approach is a backward decomposition of the variational approximation of the states posterior, combined with a representation of the ELBO and its gradients as expectations of value functions satisfying Bellman-type recursions akin to those appearing in Reinforcement Learning (RL) [39].

## 2 Background

**State-Space Models.** SSMs are defined by a latent $\mathcal{X}$-valued Markov process $(x_t)_{t \geq 1}$ and $\mathcal{Y}$-valued observations $(y_t)_{t \geq 1}$, which are conditionally independent given the latent process. They correspond to the generative model

$$x_1 \sim \mu_\theta(x_1), \qquad x_{t+1}|x_t \sim f_\theta(x_{t+1}|x_t), \qquad y_t|x_t \sim g_\theta(y_t|x_t),$$

where $\theta \in \mathbb{R}^{d_\theta}$ is a parameter of interest and we consider here $\mathcal{X} = \mathbb{R}^{d_x}$. For $y^t := y_{1:t} = (y_1, ..., y_t)$, we thus have

$$p_\theta(x_{1:t}, y^t) = \mu_\theta(x_1) g_\theta(y_1|x_1) \prod_{k=2}^{t} f_\theta(x_k|x_{k-1}) g_\theta(y_k|x_k).$$

Assume for the time being that $\theta$ is known. Given observations $(y_t)_{t \geq 1}$ and parameter values $\theta$, one can perform online state inference by computing the posterior of $x_t$ given $y^t$ which satisfies

$$p_\theta(x_t|y^t) = \frac{g_\theta(y_t|x_t) p_\theta(x_t|y^{t-1})}{p_\theta(y_t|y^{t-1})}, \quad p_\theta(x_t|y^{t-1}) = \int f_\theta(x_t|x_{t-1}) p_\theta(x_{t-1}|y^{t-1}) \mathrm{d}x_{t-1}, \quad (1)$$

with $p_\theta(x_1|y^0) := \mu_\theta(x_1)$. The log evidence $\ell_t(\theta) := \log p_\theta(y^t)$ is then given by

$$\ell_t(\theta) = \sum_{k=1}^{t} \log p_\theta(y_k|y^{k-1}), \quad \text{where} \quad p_\theta(y_k|y^{k-1}) = \int g_\theta(y_k|x_k) p_\theta(x_k|y^{k-1}) \mathrm{d}x_k. \quad (2)$$

Here $p_\theta(x_k|y^k)$ is known as the filtering distribution. While the recursion (1) and the sequential decomposition (2) are at the core of most existing online state and parameter inference techniques [19, 37, 45], we will focus here on the joint state posterior distribution, also known as the smoothing distribution, of the states $x_{1:t}$ given $y^t$ and the corresponding representation of the log evidence

$$p_\theta(x_{1:t}|y^t) = p_\theta(x_{1:t}, y^t)/p_\theta(y^t), \qquad \ell_t(\theta) = \log p_\theta(y^t) = \log \left( \int p_\theta(x_{1:t}, y^t) \mathrm{d}x_{1:t} \right).$$

**Variational Inference.** For general SSMs, the filtering and smoothing distributions as well as the log evidence are not available analytically and need to be approximated. For data $y^t$, standard variational approaches use stochastic gradient techniques to maximize the following ELBO

$$\mathcal{L}_t(\theta, \phi) := \mathbb{E}_{q^\phi(x_{1:t}|y^t)} \left[ \log \left( p_\theta(x_{1:t}, y^t)/q^\phi(x_{1:t}|y^t) \right) \right] \leq \ell_t(\theta). \quad (3)$$

Maximizing this ELBO w.r.t. the parameter $\phi$ of the variational distribution $q^\phi(x_{1:t}|y^t)$ corresponds to doing variational smoothing while maximizing it w.r.t. $\theta$ corresponds to doing parameter learning.

As the true smoothing distribution satisfies $p_\theta(x_{1:t}|y^t) = p_\theta(x_1|y^t) \prod_{k=2}^{t} p_\theta(x_k|y^t, x_{k-1})$, one typically uses $q^\phi(x_{1:t}|y^t) = q^\phi(x_1|y^t) \prod_{k=2}^{t} q^\phi(x_k|y^t, x_{k-1})$ for the variational smoothing distribution; see e.g. [2, 20, 22, 43]. However this approach is not suitable for *online* variational filtering and parameter learning. Firstly, the resulting marginal $q^\phi(x_t|y^t)$ of $q^\phi(x_{1:t}|y^t)$, approximating $p_\theta(x_t|y^t)$, is typically not available analytically. Secondly, when the new observation $y_{t+1}$ is collected, this approach would require recomputing an entirely new variational smoothing distribution with a dimension that increases with time. One can attempt to bypass this problem by restricting ourselves to $q^\phi(x_{1:t}|y^t) = q^\phi(x_1|y^1) \prod_{k=2}^{t} q^\phi(x_k|y^k, x_{k-1})$ as per [30]. However, the switch from conditioning on $y^t$ to $y^k$ prohibits learning an accurate approximation of $p_\theta(x_{1:t}|y^t)$ as this formulation does not condition on all relevant data.

# 3 Online Variational Filtering and Parameter Learning

Our online variational filtering and parameter estimation approach exploits a backward factorization of the variational smoothing distribution $q^\phi(x_{1:t}|y^t)$. The ELBO and its gradients w.r.t. $\theta$ and $\phi$ are computed in an *online* manner as $t$ increases by using a combination of ideas from dynamic programming and RL, with a computational time that remains constant at each time step. To simplify notation, henceforth we will write $q_t^\phi(x_{1:t}) = q^\phi(x_{1:t}|y^t)$.

## 3.1 Backward Decomposition of the Variational Smoothing Distribution

The key property that we will be exploiting is that $p_\theta(x_{1:t}|y^t)$ satisfies

$$p_\theta(x_{1:t}|y^t) = p_\theta(x_t|y^t) \prod_{k=1}^{t-1} p_\theta(x_k|y^k, x_{k+1}), \quad p_\theta(x_k|y^k, x_{k+1}) = \frac{f_\theta(x_{k+1}|x_k)p_\theta(x_k|y^k)}{p_\theta(x_{k+1}|y^k)}. \quad (4)$$

Equation (4) shows that, conditional upon $y^t$, $(x_k)_{k=1}^t$ is a reverse-time Markov chain of initial distribution $p_\theta(x_t|y^t)$ and backward Markov transition kernels $p_\theta(x_k|y^k, x_{k+1})$; see e.g. [12, 19]. Crucially the backward transition kernel at time $k$ depends only on the observations until time $k$.

To exploit this, we consider a variational smoothing distribution of the form

$$q_t^\phi(x_{1:t}) = q_t^\phi(x_t) \prod_{k=1}^{t-1} q_{k+1}^\phi(x_k|x_{k+1}), \quad (5)$$

where $q_t^\phi(x_t)$ and $q_{k+1}^\phi(x_k|x_{k+1})$ are variational approximations of the filtering distribution $p_\theta(x_t|y^t)$ and the backward kernel $p_\theta(x_k|y^k, x_{k+1})$ respectively. Using (5), one now has

$$\mathcal{L}_t(\theta, \phi) = \ell_t(\theta) - \mathrm{KL}(q_t^\phi(x_t)||p_\theta(x_t|y^t))$$
$$- \sum_{k=1}^{t-1} \mathbb{E}_{q_t^\phi(x_{k+1})} \left[ \mathrm{KL}(q_{k+1}^\phi(x_k|x_{k+1})||p_\theta(x_k|y^k, x_{k+1})) \right],$$

where KL is the Kullback–Leibler divergence and $q_t^\phi(x_{k+1})$ is the marginal distribution of $x_{k+1}$ under the variational distribution $q_t^\phi(x_{1:t})$ defined in (5). Considering this variational distribution thus makes it possible to learn an arbitrarily accurate variational approximation of the true smoothing distribution whilst still only needing to condition on $y^k$ at time $k$ and not on future observations. Additionally, it follows directly from (5) that we can easily update $q_{t+1}^\phi(x_{1:t+1})$ from $q_t^\phi(x_{1:t})$ using

$$q_{t+1}^\phi(x_{1:t+1}) = q_t^\phi(x_{1:t})m_{t+1}^\phi(x_{t+1}|x_t), \text{ for } m_{t+1}^\phi(x_{t+1}|x_t) := \frac{q_{t+1}^\phi(x_t|x_{t+1})q_{t+1}^\phi(x_{t+1})}{q_t^\phi(x_t)}. \quad (6)$$

Here $m_{t+1}^\phi(x_{t+1}|x_t)$ can be viewed as an approximation of the Markov transition density $q_{t+1}^\phi(x_{t+1}|x_t) \propto q_{t+1}^\phi(x_t|x_{t+1})q_{t+1}^\phi(x_{t+1})$ but it is typically not a proper Markov transition density; i.e. $\int m_{t+1}^\phi(x_{t+1}|x_t)\mathrm{d}x_{t+1} \neq 1$ as $\int q_{t+1}^\phi(x_t|x_{t+1})q_{t+1}^\phi(x_{t+1})\mathrm{d}x_{t+1} \neq q_t^\phi(x_t)$.

Let us assume that $q_k^\phi(x_k) = q_k^{\phi_k}(x_k)$ and $q_k^\phi(x_{k-1}|x_k) = q_k^{\phi_k}(x_{k-1}|x_k)$, then (5) and (6) suggest that we only need to estimate $\phi_t$ at time $t$ as $y_t$ does not impact the backward Markov kernels prior to time $t$. However, we also have to be able to compute estimates of $\nabla_\phi \mathcal{L}_t(\theta, \phi)$ and $\nabla_\theta \mathcal{L}_t(\theta, \phi)$ to optimize parameters in a constant computational time at each time step, without having to consider the entire history of observations $y^t$. This is detailed in the next subsections where we show that the sequence of ELBOs $\{\mathcal{L}_t(\theta, \phi)\}_{t \geq 1}$ and its gradients $\{\nabla_\theta \mathcal{L}_t(\theta, \phi)\}_{t \geq 1}$ and $\{\nabla_\phi \mathcal{L}_t(\theta, \phi)\}_{t \geq 1}$ can all be computed online when using the variational distributions $\{q_t^\phi(x_{1:t})\}_{t \geq 1}$ defined in (5).

## 3.2 Forward recursion for the ELBO

We start by presenting a forward-only recursion for the computation of $\{\mathcal{L}_t(\theta, \phi)\}_{t \geq 1}$. This recursion illustrates the parallels between variational inference and RL and is introduced to build intuition.

**Proposition 1.** *The ELBO $\mathcal{L}_t(\theta, \phi)$ satisfies for $t \geq 1$*

$$\mathcal{L}_t(\theta, \phi) = \mathbb{E}_{q_t^\phi(x_t)}[V_t^{\theta, \phi}(x_t)] \quad for \quad V_t^{\theta, \phi}(x_t) := \mathbb{E}_{q_t^\phi(x_{1:t-1}|x_t)} \left[ \log \left( p_\theta(x_{1:t}, y^t)/q_t^\phi(x_{1:t}) \right) \right],$$

*with the convention $V_1^{\theta,\phi}(x_1) := r_1^{\theta,\phi}(x_0, x_1) := \log(p_\theta(x_1, y_1)/q_1^\phi(x_1))$. Additionally, we have*

$$V_{t+1}^{\theta,\phi}(x_{t+1}) = \mathbb{E}_{q_{t+1}^\phi(x_t|x_{t+1})}[V_t^{\theta,\phi}(x_t) + r_{t+1}^{\theta,\phi}(x_t, x_{t+1})], \quad \text{where} \tag{7}$$

$$r_{t+1}^{\theta,\phi}(x_t, x_{t+1}) := \log\left(f_\theta(x_{t+1}|x_t)g_\theta(y_{t+1}|x_{t+1})/m_{t+1}^\phi(x_{t+1}|x_t)\right). \tag{8}$$

All proofs are given in Appendix C. Proposition 1 shows that we can compute $\mathcal{L}_t(\theta, \phi)$, for $t \geq 1$, online by recursively computing the functions $V_t^{\theta,\phi}$ using (7) and then taking the expectation of $V_t^{\theta,\phi}$ w.r.t. $q_t^\phi(x_t)$ to obtain the ELBO at time $t$. Thus, given $V_t^{\theta,\phi}$, we can compute $V_{t+1}^{\theta,\phi}$ and $\mathcal{L}_{t+1}(\theta, \phi)$ using only $y_{t+1}$, with a cost that remains constant in $t$.

This type of recursion is somewhat similar to those appearing in RL. We can indeed think of the ELBO $\mathcal{L}_t(\theta, \phi)$ as the expectation of a sum of "rewards" $r_k^{\theta,\phi}$ given in (8) from $k = 1$ to $k = t$ which we compute recursively using the "value" function $V_t^{\theta,\phi}$. However, while in RL the value function is given by the expectation of the sum of future rewards starting from $x_t$, the value function defined here is the expectation of the sum of past rewards conditional upon arriving in $x_t$. This yields the required forward recursion instead of a backward recursion. We expand on this link in Section 4 and Appendix A.3.

### 3.3 Forward recursion for ELBO gradient w.r.t. $\theta$

A similar recursion can be obtained to compute $\{\nabla_\theta \mathcal{L}_t(\theta, \phi)\}_{t \geq 1}$. This recursion will be at the core of our online parameter learning algorithm. Henceforth, we will assume that regularity conditions allowing both differentiation and the interchange of integration and differentiation are satisfied.

**Proposition 2.** *The ELBO gradient $\nabla_\theta \mathcal{L}_t(\theta, \phi)$ satisfies for $t \geq 1$*

$$\nabla_\theta \mathcal{L}_t(\theta, \phi) = \mathbb{E}_{q_t^\phi(x_t)}[S_t^{\theta,\phi}(x_t)], \quad \text{where} \quad S_t^{\theta,\phi}(x_t) := \nabla_\theta V_t^{\theta,\phi}(x_t).$$

*Additionally, if we define $s_{t+1}^\theta(x_t, x_{t+1}) := \nabla_\theta r_{t+1}^{\theta,\phi}(x_t, x_{t+1}) = \nabla_\theta \log f_\theta(x_{t+1}|x_t)g_\theta(y_{t+1}|x_{t+1})$ then*

$$S_{t+1}^{\theta,\phi}(x_{t+1}) = \mathbb{E}_{q_{t+1}^\phi(x_t|x_{t+1})}\left[S_t^{\theta,\phi}(x_t) + s_{t+1}^\theta(x_t, x_{t+1})\right]. \tag{9}$$

Proposition 2 shows that we can compute $\{\nabla_\theta \mathcal{L}_t(\theta, \phi)\}_{t \geq 1}$ online by propagating $\{S_t^{\theta,\phi}\}_{t \geq 1}$ using (9) and taking the expectation of the vector $S_t^{\theta,\phi}$ w.r.t. $q_t^\phi(x_t)$ to obtain the gradient at time $t$. Similar ideas have been previously exploited in the statistics literature to obtain a forward recursion for the score vector $\nabla_\theta \ell_t(\theta)$ so as to perform recursive maximum likelihood parameter estimation; see e.g. [19, Section 4]. In this case, one has $\nabla_\theta \ell_t(\theta) = \mathbb{E}_{p_\theta(x_t|y^t)}[S_t^\theta(x_t)]$ where $S_t^\theta$ satisfies a recursion similar to (9) with $q_{t+1}^\phi(x_t|x_{t+1})$ replaced by $p_\theta(x_t|y^t, x_{t+1})$.

### 3.4 Forward recursion for ELBO gradient w.r.t. $\phi$

We finally establish forward-only recursions for the gradient of the ELBO w.r.t. $\phi$ which will allow us to perform online variational filtering. We consider here the case where, for all $k$, $q_k^\phi(x_k) = q_k^{\phi_k}(x_k)$ and $q_k^\phi(x_{k-1}|x_k) = q_k^{\phi_k}(x_{k-1}|x_k)$ so $\mathcal{L}_t(\theta, \phi) = \mathcal{L}_t(\theta, \phi_{1:t})$ and $V_t^{\theta,\phi}(x_t) = V_t^{\theta,\phi_{1:t}}(x_t)$. At time step $t$, we will optimize w.r.t. $\phi_t$ and hold all previous $\phi_{1:t-1}$ constant. Our overall variational posterior (5) is denoted $q_t^{\phi_{1:t}}(x_{1:t})$. The alternative approach where $\phi$ is shared through time (amortization) is investigated in Appendix D.

Since the expectation is taken w.r.t. $q_t^{\phi_{1:t}}(x_{1:t})$ in $\mathcal{L}_t$, optimizing $\phi_t$ is slightly more difficult than for $\theta$. However, we can still derive a forward recursion for the $\phi$ gradients and we will leverage the reparameterization trick to reduce the variance of the gradient estimates; i.e. we assume that $x_t(\phi_t; \epsilon_t) \sim q_t^{\phi_t}(x_t)$ and $x_{t-1}(\phi_t; \epsilon_{t-1}, x_t) \sim q_t^{\phi_t}(x_{t-1}|x_t)$ when $\epsilon_{t-1} \sim \lambda(\epsilon)$, $\epsilon_t \sim \lambda(\epsilon)$.

**Proposition 3.** *The ELBO gradient $\nabla_{\phi_t} \mathcal{L}_t(\theta, \phi_{1:t})$ satisfies for $t \geq 1$*

$$\nabla_{\phi_t} \mathcal{L}_t(\theta, \phi_{1:t}) = \nabla_{\phi_t} \mathbb{E}_{q_t^{\phi_t}(x_t)}[V_t^{\theta,\phi_{1:t}}(x_t)] = \mathbb{E}_{\lambda(\epsilon)}[\nabla_{\phi_t} V_t^{\theta,\phi_{1:t}}(x_t(\phi_t; \epsilon_t))].$$

*Additionally, one has*

$$\nabla_{\phi_{t+1}} V_{t+1}^{\theta,\phi_{1:t+1}}(x_{t+1}(\phi_{t+1};\epsilon_{t+1}))$$

$$= \mathbb{E}_{\lambda(\epsilon_t)} \left[ T_t^{\theta,\phi_{1:t}}(x_t(\phi_{t+1};\epsilon_t, x_{t+1}(\phi_{t+1};\epsilon_{t+1}))) \frac{\mathrm{d}x_t(\phi_{t+1};\epsilon_t, x_{t+1}(\phi_{t+1};\epsilon_{t+1}))}{\mathrm{d}\phi_{t+1}} \right.$$

$$\left. + \nabla_{\phi_{t+1}} r_{t+1}^{\theta,\phi_{t:t+1}}(x_t(\phi_{t+1};\epsilon_t, x_{t+1}(\phi_{t+1};\epsilon_{t+1})), x_{t+1}(\phi_{t+1};\epsilon_{t+1})) \right],$$

*where* $T_t^{\theta,\phi_{1:t}}(x_t) := \frac{\partial}{\partial x_t} V_t^{\theta,\phi_{1:t}}(x_t)$ *satisfies the forward recursion*

$$T_{t+1}^{\theta,\phi_{1:t+1}}(x_{t+1}) = \mathbb{E}_{\lambda(\epsilon_t)} \left[ T_t^{\theta,\phi_{1:t}}(x_t(\phi_{t+1};\epsilon_t, x_{t+1})) \frac{\partial x_t(\phi_{t+1};\epsilon_t, x_{t+1})}{\partial x_{t+1}} \right.$$

$$\left. + \nabla_{x_{t+1}} r_{t+1}^{\theta,\phi_{t:t+1}}(x_t(\phi_{t+1};\epsilon_t, x_{t+1}), x_{t+1}) \right]. \quad (10)$$

*Here,* $\frac{\mathrm{d}x_t(\phi_{t+1};\epsilon_t, x_{t+1}(\phi_{t+1};\epsilon_{t+1}))}{\mathrm{d}\phi_{t+1}}, \frac{\partial x_t(\phi_{t+1};\epsilon_t, x_{t+1})}{\partial x_{t+1}}$ *are Jacobians of appropriate dimensions.*

### 3.5 Estimating the ELBO and its Gradients

As we consider the case $q_k^\phi(x_k) = q_k^{\phi_k}(x_k)$, $q_k^\phi(x_{k-1}|x_k) = q_k^{\phi_k}(x_{k-1}|x_k)$, we have $S_t^{\theta,\phi}(x_t) = S_t^{\theta,\phi_{1:t}}(x_t)$. At time $t$ we optimize $\phi_t$ and hold $\phi_{1:t-1}$ constant. Practically, we are not able to compute in closed-form the functions $V_t^{\theta,\phi_{1:t}}(x_t)$, $S_t^{\theta,\phi_{1:t}}(x_t)$ and $T_t^{\theta,\phi_{1:t}}(x_t)$ appearing in the forward recursions of $\mathcal{L}_t(\theta,\phi_{1:t})$, $\nabla_\theta \mathcal{L}_t(\theta,\phi_{1:t})$ and $\nabla_\phi \mathcal{L}_t(\theta,\phi_{1:t})$ respectively. However, we can exploit the above recursions to approximate these functions online using regression as is commonly done in RL. We then show how to use these gradients for online filtering and parameter learning.

We approximate $S_{t+1}^{\theta,\phi_{1:t+1}}$ with $\hat{S}_{t+1}$. Equation (9) shows that $\hat{S}_{t+1}$ can be learned using $\hat{S}_t$ through regression of the simulated dataset[2] $\{x_{t+1}^i, \hat{S}_t(x_t^i) + s_{t+1}^\theta(x_t^i, x_{t+1}^i)\}$ with $(x_t^i, x_{t+1}^i) \overset{\text{i.i.d.}}{\sim} q_{t+1}^{\phi_{t+1}}(x_t, x_{t+1})$ for $i = 1, ..., N$ (see Appendix A.1 for derivation). We can use neural networks to model $\hat{S}_t$ or Kernel Ridge Regression (KRR). Note that the use of KRR to estimate gradients for variational learning has recently been demonstrated by [26].

We similarly approximate $T_{t+1}^{\theta,\phi_{1:t+1}}$ with $\hat{T}_{t+1}$. As before, we can model $\hat{T}_{t+1}$ using neural networks or KRR. We use recursion (10) and $\hat{T}_t$ to create the following dataset for regression[3]

$$\left\{ x_{t+1}^i, \ \hat{T}_t(x_t(\phi_{t+1};\epsilon_t^i, x_{t+1}^i)) \frac{\partial x_t(\phi_{t+1};\epsilon_t^i, x_{t+1}^i)}{\partial x_{t+1}^i} + \nabla_{x_{t+1}^i} r_{t+1}^{\theta,\phi_{t:t+1}}(x_t(\phi_{t+1};\epsilon_t^i, x_{t+1}^i), x_{t+1}^i) \right\},$$

where $x_{t+1}^i \sim q_{t+1}^{\phi_{t+1}}(x_{t+1})$ and $\epsilon_t^i \sim \lambda(\epsilon)$ for $i = 1, ..., N$. The choice of the distribution over the inputs, $x_{t+1}^i$, for each simulated dataset is arbitrary but it will determine where the approximations are most accurate. We choose $q_{t+1}^{\phi_{t+1}}(x_{t+1})$ to best match where we expect the approximations to be evaluated, more details are given in Appendix A.1.

Note that if one is interested in computing online an approximation of the ELBO, we can again similarly approximate $V_t^{\theta,\phi_{1:t}}(x_t)$ using regression to obtain $\hat{V}_t(x_t)$ by leveraging (7). We will call the resulting approximate ELBO the Recursive ELBO (RELBO). We could also then differentiate $\hat{V}_t(x_t)$ w.r.t. $x_t$ to obtain an alternative method for estimating $T_t^{\theta,\phi_{1:t}}$ and optimizing $\phi_t$. However, as we are ultimately interested in accurate gradients, this approach does not exploit the readily available gradient information during the regression stage.

By approximating $T_{t+1}^{\theta,\phi_{1:t+1}}$ with $\hat{T}_{t+1}$ and $S_{t+1}^{\theta,\phi_{1:t+1}}$ with $\hat{S}_{t+1}$, we introduce some bias into our gradient estimates. We can trade bias for variance by using modified recursions; e.g.

$$S_{t+1}^{\theta,\phi_{1:t+1}}(x_{t+1}) = \mathbb{E}_{q_t^{\phi_{t-L+2:t+1}}(x_{t-L+1:t}|x_{t+1})} \left[ S_{t-L+1}^{\theta,\phi_{1:t-L+1}}(x_{t-L+1}) + \sum_{k=t-L+1}^t s_{k+1}^\theta(x_k, x_{k+1}) \right].$$

As $L$ increases, we will reduce bias but increase variance. Such ideas are also commonly used in RL but we will limit ourselves here to using $L = 1$.

---

[2] We define $\hat{S}_0 := 0$, $x_0^i := \emptyset$, $s_1^\theta(x_0, x_1) = \nabla_\theta \log \mu_\theta(x_1) g_\theta(y_1|x_1)$ and $q_1^{\phi_1}(x_0, x_1) := q_1^{\phi_1}(x_1)$.

[3] Similarly, we define $\hat{T}_0 := 0$, $x_0^i := \emptyset$ and $r_1^{\theta,\phi_{0:1}}$ as in Proposition 1.

---

**Algorithm 1:** Online Variational Filtering and Parameter Learning.

---

**for** $t = 1, \dots, T$ **do**

    Initialize $\phi_t$ e.g. $\phi_t \leftarrow \phi_{t-1}$

    `/* Update `$\phi_t$` using `$M$` stochastic gradient steps`             `*/`

    **for** $m = 1, \dots, M$ **do**

        Sample $x_{t-1}^i, x_t^i \sim q_t^{\phi_t}(x_{t-1}, x_t)$ using reparameterization trick for $i = 1, \dots, N$

        $\phi_t \leftarrow \phi_t + \gamma_m \frac{1}{N} \sum_{i=1}^N \{ \hat{T}_{t-1}(x_{t-1}^i) \frac{\mathrm{d}x_{t-1}^i}{\mathrm{d}\phi_t} + \nabla_{\phi_t} r_t(x_{t-1}^i, x_t^i) \}$

    **end**

    `/* Update `$\hat{T}_t(x_t)$` and `$\hat{S}_t(x_t)$` as in Section `3.5              `*/`

    $\hat{T}_t(x_t) \overset{regression}{\leftarrow} \hat{T}_{t-1}(x_{t-1}) \frac{\partial x_{t-1}}{\partial x_t} + \nabla_{x_t} r_t(x_{t-1}, x_t).$

    $\hat{S}_t(x_t) \overset{regression}{\leftarrow} \hat{S}_{t-1}(x_{t-1}) + s_t^{\theta_{t-1}}(x_{t-1}, x_t).$

    `/* Update `$\theta$` using a stochastic gradient step`              `*/`

    Sample $x_{t-1}^i, x_t^i \sim q_t^{\phi_t}(x_{t-1}, x_t), \quad \tilde{x}_{t-1}^i \sim q_t^{\phi_{t-1}}(x_{t-1})$ for $i = 1, \dots, N$

    $\theta_t \leftarrow \theta_{t-1} + \eta_t \frac{1}{N} \sum_{i=1}^N \{ \hat{S}_{t-1}(x_{t-1}^i) + s_t^{\theta_{t-1}}(x_{t-1}^i, x_t^i) - \hat{S}_{t-1}(\tilde{x}_{t-1}^i) \}$

**end**

---

### 3.6 Online Parameter Estimation

Assume, for the sake of argument, that one has access to the log evidence $\ell_t(\theta)$ and that the observations arise from the SSM with parameter $\theta^\star$. Under regularity conditions, the average log-likelihood $\ell_t(\theta)/t$ converges as $t \to \infty$ towards a function $\ell(\theta)$ which is maximized at $\theta^\star$; see e.g. [12, 41]. We can maximize this criterion online using Recursive Maximum Likelihood Estimation (RMLE) [19, 24, 40, 41] which consists of updating the parameter estimate $\theta$ using

$$\theta_{t+1} = \theta_t + \eta_{t+1} \left( \mathbb{E}_{p_{\theta_{0:t}}(x_t, x_{t+1}|y^{t+1})}[S_t(x_t) + s_{t+1}^{\theta_t}(x_t, x_{t+1})] - \mathbb{E}_{p_{\theta_{0:t-1}}(x_t|y^t)}[S_t(x_t)] \right). \quad (11)$$

The difference of two expectations on the r.h.s. of (11) is an approximation of the gradient of the log-predictive $\log p_\theta(y_{t+1}|y^t)$ evaluated at $\theta_t$. The approximation is given by $\nabla \log p_{\theta_{0:t}}(y^{t+1}) - \nabla \log p_{\theta_{0:t-1}}(y^t)$ with the notation $\nabla \log p_{\theta_{0:t}}(y^{t+1})$ corresponding to the expectation of the sum of terms $s_{k+1}^{\theta_k}(x_k, x_{k+1})$ w.r.t. the joint posterior states distribution defined by using the SSM with parameter $\theta_k$ at time $k+1$.

We proceed similarly in the variational context and update the parameter using

$$\theta_{t+1} = \theta_t + \eta_{t+1} \left( \mathbb{E}_{q_{t+1}^{\phi_{t+1}}(x_t, x_{t+1})} \left[ \hat{S}_t(x_t) + s_{t+1}^{\theta_t}(x_t, x_{t+1}) \right] - \mathbb{E}_{q_t^{\phi_t}(x_t)} \left[ \hat{S}_t(x_t) \right] \right).$$

Here $\hat{S}_t(x_t)$ approximates $S_t(x_t)$ satisfying $S_{t+1}(x_{t+1}) := \mathbb{E}_{q_{t+1}^{\phi_{t+1}}(x_t|x_{t+1})}[S_t(x_t) + s_{t+1}^{\theta_t}(x_t, x_{t+1})]$.

We compute $\hat{S}_t$ as in Section 3.5 with a simulated dataset using $\hat{S}_{t-1}$ and $\theta_{t-1}$.

Putting everything together, we summarize our method in Algorithm 1 using a simplified notation to help build intuition, a full description is given in Appendix A.2. It is initialized using initial parameters $\phi_1, \theta_0$. We re-iterate the algorithm's computational cost does not grow with $t$. We need only store fixed size $\hat{T}$ and $\hat{S}$ models as well as the most recent $\phi_t$ and $\theta_t$ parameters. When performing backpropagation, $\hat{T}$ and $\hat{S}$ summarize all previous gradients, meaning we do not have to roll all the way back to $t = 1$. Therefore, we only incur standard backpropagation computational cost w.r.t. $\phi_t$ and $\theta$. To scale to large $d_x$, we can use mean field $q_t^{\phi_t}(x_t)$, $q_t^{\phi_t}(x_{t-1}|x_t)$ keeping costs linear in $d_x$.

## 4 Related Work

As mentioned briefly in Section 3.2, our recursive method has close links with RL, which we make explicit here. In 'standard' RL, we have a value function and corresponding Bellman recursion given by

$$V_t^{\text{RL}}(s_t) := \mathbb{E} \left[ \sum_{k=t}^T r(s_k, a_k) \right], \qquad V_t^{\text{RL}}(s_t) = \mathbb{E}_{s_{t+1}, a_t} \left[ r(s_t, a_t) + V_{t+1}^{\text{RL}}(s_{t+1}) \right], \qquad (12)$$

where $(s_t, a_t)$ is the state-action pair at time $t$. Whereas the RL value function summarizes future rewards and so recurses backward in time, our 'value' function summarizes previous rewards and recurses forward in time. Writing this using the state-action notation, one obtains

$$V_t(s_t) := \mathbb{E}\left[\sum_{k=1}^{t} r(s_k, a_k)\right], \qquad V_{t+1}(s_{t+1}) = \mathbb{E}_{a_{t+1}, s_t}\left[r(s_{t+1}, a_{t+1}) + V_t(s_t)\right]. \qquad (13)$$

Note we have defined an *anti-causal* graphical model, with $s_t$ depending on $s_{t+1}$ and $a_{t+1}$. If we further let $s_t = x_t$, $a_t = x_{t-1} \sim q_t^{\phi}(x_{t-1}|x_t)$, $P(s_t|s_{t+1}, a_{t+1}) = \delta(s_t = a_{t+1})$ and $r(s_t, a_t) = r(x_t, x_{t-1})$ as in equation (8), then the recursion in equation (13) for $V_t$ corresponds to the recursion in equation (7) and $\mathcal{L}_T = \mathbb{E}_{q_T^{\phi}(x_T)}[V_T(x_T)]$. We then differentiate this recursion with respect to $\theta$ and $x_{t+1}$ to get our gradient recursions (9) and (10) respectively. Full details are given in Appendix A.3. A similar correspondence between state actions and successive hidden states was also noted in [43] which explores the use of RL ideas in the context of variational inference. However, [43] exploits this correspondence to propose a backward in time recursion for the ELBO of the form (12) initialized at the time of the last observation of the SSM. When a new observation is collected at the next time step, this backward recursion has to be re-run which would lead to a computational time increasing linearly at each time step. In [20], the links between RL and variational inference are also exploited. The likelihood of future points in an SSM is approximated using temporal difference learning [39], but the proposed algorithm is not online.

To perform online variational inference, [45] proposes to use the decomposition (2) of the log evidence $\ell_t(\theta)$ and lower bound each term $\log p_\theta(y_k|y^{k-1})$ appearing in the sum using

$$\mathbb{E}_{q_k^{\phi}(x_{k-1}, x_k)}\left[\log \frac{f_\theta(x_k|x_{k-1})g_\theta(y_k|x_k)p_\theta(x_{k-1}|y^{k-1})}{q_k^{\phi}(x_{k-1}, x_k)}\right] \leq \log p_\theta(y_k|y^{k-1}). \qquad (14)$$

Unfortunately, the term on the l.h.s. of (14) cannot be evaluated unbiasedly as it relies on the intractable filter $p_\theta(x_{k-1}|y^{k-1})$ so [45] approximates it by $p_\theta(x_{k-1}|y^{k-1}) \approx q_{k-1}^{\phi}(x_{k-1})$ to obtain the following *Approximate* ELBO (AELBO) by summing over $k$:

$$\widetilde{\mathcal{L}}_t(\theta, \phi_{1:t}) = \sum_{k=1}^{t} \mathbb{E}_{q_k^{\phi}(x_{k-1}, x_k)}\left[\log r_k^{\theta, \phi}(x_{k-1}, x_k)\right]. \qquad (15)$$

Additionally, [45] makes the assumption $q_k^{\phi}(x_{k-1}, x_k) = q_{k-1}^{\phi}(x_{k-1})q_k^{\phi}(x_k)$ and we will refer to (15) in this case as AELBO-1. While [16] does not consider online learning, their objective is actually a generalization of [45], with $q_k^{\phi}(x_{k-1}, x_k) = q_k^{\phi}(x_k)q_k^{\phi}(x_{k-1}|x_k)$, and we will refer to (15) in this case as AELBO-2. It can be easily shown that AELBO-2 is only equal to the true ELBO given in Proposition 1 in the (unrealistic) scenario where $q_k^{\phi}(x_k) = p_\theta(x_k|y^k)$ for all $k$. Moreover, in both cases the term $p_\theta(x_{k-1}|y^{k-1})$ is replaced by $q_{k-1}^{\phi}(x_{k-1})$, causing a term involving $\theta$ to be ignored in gradient computation. The approach developed here can be thought of as a way to correct the approximate ELBOs computed in [16, 45] in a principled manner, which takes into account the discrepancy between the filtering and approximate filtering distributions, and maintains the correct gradient dependencies in the computation graph. Finally [44] relies on the PF to do online variational inference. However the variational approximation of the filtering distribution is only implicit as its expression includes an intractable expectation and, like any other PF technique, its performance is expected to degrade significantly with the state dimension [6].

## 5 Experiments

### 5.1 Linear Gaussian State-Space Models

We first consider a linear Gaussian SSM for which the filtering distributions can be computed using the KF and the RMLE recursion (11) can be implemented exactly. Here the model is defined as

$$f_\theta(x_t|x_{t-1}) = \mathcal{N}(x_t; Fx_{t-1}, U), \quad g_\theta(y_t|x_t) = \mathcal{N}(y_t; Gx_t, V),$$

where $F \in \mathbb{R}^{d_x \times d_x}$, $G \in \mathbb{R}^{d_y \times d_x}$, $U \in \mathbb{R}^{d_x \times d_x}$, $V \in \mathbb{R}^{d_y \times d_y}$, $\theta = \{F, G\}$. We let

$$q_t^{\phi_t}(x_t) = \mathcal{N}\left(x_t; \mu_t, \text{diag}(\sigma_t^2)\right), \quad q_t^{\phi_t}(x_{t-1}|x_t) = \mathcal{N}\left(x_{t-1}; W_t x_t + b_t, \text{diag}(\tilde{\sigma}_t^2)\right),$$

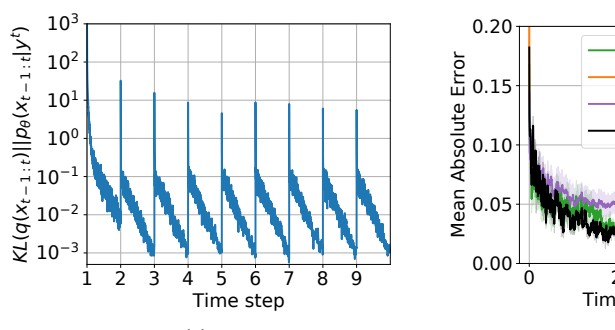
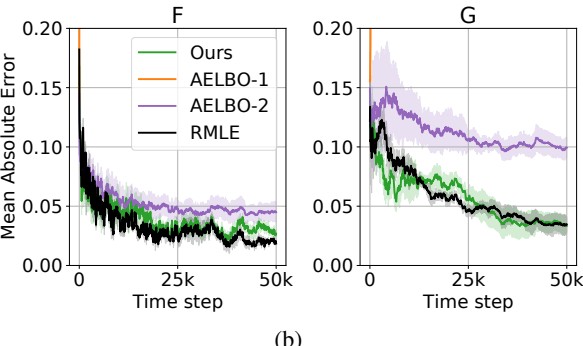

Figure 1: **(a)** $\text{KL}(q_t^{\phi_t}(x_{t-1}, x_t) || p_\theta(x_{t-1}, x_t | y^t))$ vs time step of the SSM. Between each time step, we plot the progress of the KL over 5000 iterations of inner loop $\phi_t$ optimization. **(b)** Mean Absolute Error for model parameters $F$ (left) and $G$ (right) vs time step (AELBO-1 off the scale).

with $\phi_t = \{\mu_t, \log \sigma_t, W_t, b_t, \log \tilde{\sigma}_t\}$. When $f_\theta(x_t | x_{t-1})$ is linear Gaussian and we use a Gaussian for $q_{t-1}^{\phi_{t-1}}(x_{t-1})$, we can select $q_t^{\phi_t}(x_{t-1}|x_t) \propto f_\theta(x_t|x_{t-1}) q_{t-1}^{\phi_{t-1}}(x_{t-1})$ as noted in [34]. In this experiment however, we aim to show our full method can converge to known ground truth values hence still fully parameterize $q_t^{\phi_t}(x_{t-1}|x_t)$ as well as setting the matrices $F, G, U, V$ to be diagonal, so that $p_\theta(x_t|y^t)$ and $p_\theta(x_{t-1}|y^{t-1}, x_t)$ are in the variational family.

For $d_x = d_y = 10$, we first demonstrate accurate state inference by learning $\phi_t$ at each time step whilst holding $\theta$ fixed at the true value. We represent $\hat{T}_t(x_t)$ non-parametrically using KRR. Full details for all experiments are given in Appendix B.[4] Figure 1a illustrates how, given extra computation, our variational approximation comes closer and closer to the ground truth, the accuracy being limited by the convergence of each inner stochastic gradient ascent procedure. We then consider online learning of the parameters $F$ and $G$ using Algorithm 1, comparing our result to RMLE and a variation of Algorithm 1 using AELBO-1 and 2 (see Section 4). Our methodology converges much closer to the analytic baseline (RMLE) than AELBO-2 [16] and exhibits less variance, even though the variational family is sufficiently expressive for AELBO-2 to learn the correct backward transition. In addition, we find that AELBO-1 [45] did not produce reliable parameter estimates in this example, as it relies on a variational approximation that ignores the dependence between $x_{k-1}$ and $x_k$. As expected, our method performs slightly worse than the analytic RMLE, as inevitably small errors will be introduced during stochastic optimization and regression.

## 5.2 Chaotic Recurrent Neural Network

We next evaluate the performance of our algorithm for state estimation in non-linear, high-dimensional SSMs. We reproduce the Chaotic Recurrent Neural Network (CRNN) example in [44], but with state dimension $d_x = 5, 20$, and 100. This non-linear model is an Euler approximation of the continuous-time recurrent neural network dynamics

$$f(x_t|x_{t-1}) = \mathcal{N}\left(x_t; x_{t-1} + \Delta\tau^{-1}\left(-x_{t-1} + \gamma W \tanh(x_{t-1})\right), U\right),$$

and the observation model is linear with additive noise from a Student's t-distribution. We compare our algorithm against ensemble KF (EnKF), bootstrap PF (BPF), as well as variational methods using AELBO-1 and AELBO-2. We let $q_t^{\phi_t}(x_{t-1}|x_t) = \mathcal{N}(x_{t-1}; \text{MLP}_t^{\phi_t}(x_t), \text{diag}(\tilde{\sigma}_t^2))$ and $q_t^{\phi_t}(x_t) = \mathcal{N}\left(x_t; \mu_t, \text{diag}(\sigma_t^2)\right)$ where we use a 1-layer Multi-Layer Perceptron (MLP) with 100 neurons for each $q_t^{\phi_t}(x_{t-1}|x_t)$. We generate a dataset of length 100 using the same settings as [44], and each algorithm is run 10 times to report the mean and standard deviation. We also match approximately the computational complexity for all methods. From Table 1, we observe that the EnKF performs poorly on this non-linear model, while the PF performance degrades significantly with $d_x$, as expected. Among variational methods, AELBO-1 does not give as accurate state estimation, while AELBO-2 and our method achieve the lowest error in terms of RMSE. However, our method achieves the highest ELBO; i.e. lowest KL between the variational approximation and the true posterior since

---

[4]Code available at https://github.com/andrew-cr/online_var_fil

$\theta$ is fixed - an effect not represented using just the RMSE. We confirm this is the case in Appendix B.2 by comparing our variational filter means $\mu_t$ against the 'ground truth' posterior mean for $d_x = 5$ computed using PF with 10 million particles. Furthermore, our method is also able to accurately estimate the true ELBO online. Figure 2a shows that our online estimate of the ELBO, RELBO (Section 3.5), is very close to the true ELBO, whereas AELBO-2 is biased and consistently overestimates it. Further, AELBO-1 is extremely loose meaning its approximation of the joint state posterior is very poor.

Using this CRNN problem, we also investigate the penalty we pay for our online method. At time $t$, we train $q_t^{\phi_t}(x_t)$, $q_t^{\phi_t}(x_{t-1}|x_t)$ and hold $q_k^{\phi_k}(x_{k-1}|x_k)$, $k < t$ fixed. This is because all information to learn the true factor $p_\theta(x_{k-1}|y^{k-1}, x_k)$ is available at time $k$ as it does not depend on future observations. However, when $p_\theta(x_{k-1}|y^{k-1}, x_k)$ is not in the variational family (as in this CRNN example), $q_k^{\phi_k}(x_{k-1}|x_k)$ will aim to be most accurate in the regions of $x_k$ in which it is typically evaluated. The evaluation distribution over $x_k$ does depend on future variational factors and so could shift over time. This may result in learning a different variational joint $q_t^{\phi_{1:t}}(x_{1:t})$ between when using our online ELBO and when just using the full offline ELBO (3). We quantify this difference by training the same joint variational distribution using either our online ELBO or the offline ELBO (3). Figure 2b plots the final mean and standard deviations of the marginals of the trained $q_t^{\phi_{1:t}}(x_{1:t})$ in both cases. We see that these quantities are very close, suggesting this effect is not an issue on this example. This may be due to the evaluation distribution over $x_k$ not changing a significant enough amount to cause appreciable changes in the learned variational factors. We show this result holds over dimensions and seeds in Appendix B.2.

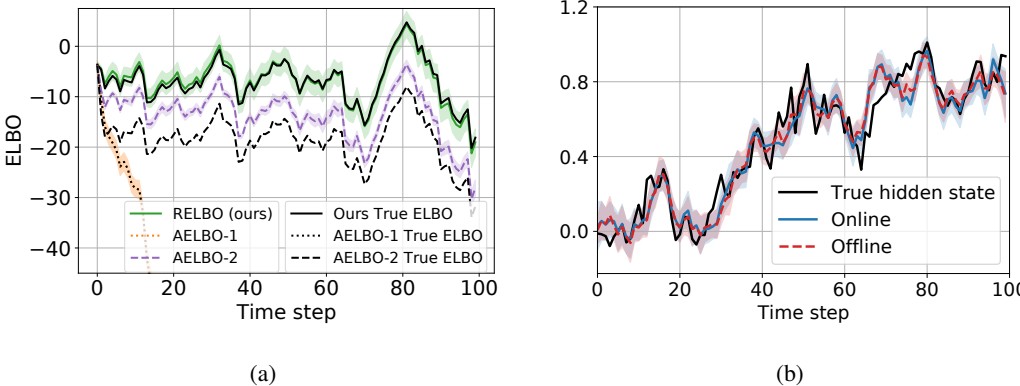

(a)                                                          (b)

Figure 2: **(a)** Estimates and true values of the ELBO on the Chaotic RNN task. RELBO uses KRR for $\hat{V}_t$ whilst for the other methods we use eq. (15). **(b)** Comparison between joint variational distributions trained online and offline on the CRNN task. The colored lines show the mean of $q_t^{\phi_{1:t}}(x_{1:t})$ whilst the shaded region represent $\pm 1$ std. The true hidden state is also shown in black.

### 5.3 Sequential Variational Auto-Encoder

We demonstrate the scalability of our method on a sequential VAE application. In this problem, an agent observes a long sequence of frames that could, for example, come from a robot traversing a new environment. The frames are encoded into a latent representation using a pre-trained decoder. The agent must then learn online the transition dynamics within this latent space using the single stream of input images. The SSM is defined by

$$f_\theta(x_t|x_{t-1}) = \mathcal{N}(x_t; \text{NN}_\theta^f(x_{t-1}), U), \qquad g(y_t|x_t) = \mathcal{N}(y_t; \text{NN}^g(x_t), V),$$

where $d_x = 32$, $\text{NN}_\theta^f$ is a residual MLP and $\text{NN}^g$ a convolutional neural network. $\text{NN}_\theta^f$ is learned online whilst $\text{NN}^g$ is fixed and is pre-trained on unordered images from similar environments using the standard VAE objective [21]. We perform this experiment on a video sequence from a DeepMind Lab environment [5] (GNU GPL license). We use the same $q_t^{\phi_t}$ parameterization as for the CRNN but with a 2 hidden layer MLP with 64 neurons. KRR is used to learn $\hat{T}_t$ whereas we use an MLP for learning $\hat{S}_t$. We found that MLPs scale better than KRR as $d_\theta$ is high. Our online algorithm is run on a sequence of 4000 images after which we probe the quality of the learned $\text{NN}_\theta^f$. The results are

Table 1: Root Mean Squared Error between filtering mean and true state and the average true ELBO for the 5 methods in varying dimensions on the Chaotic RNN task.

| $d_x$ | | EnKF | BPF | AELBO-1 | AELBO-2 | Ours |
|---|---|---|---|---|---|---|
| 5 | Filter RMSE | 0.1450±0.0026 | **0.1026±0.0001** | 0.1284±0.0035 | 0.1035±0.0012 | 0.1032±0.0005 |
| | ELBO (nats) | - | - | -220.52±6.2768 | -30.944±2.2928 | **-15.845±1.7385** |
| | Time per step | 1.0998 | 0.9268 | 1.5067 | 2.2270 | 2.6899 |
| 20 | Filter RMSE | 0.1541±0.0016 | 0.1092±0.0014 | 0.1355±0.0012 | 0.1086±0.0004 | **0.1082±0.0003** |
| | ELBO (nats) | - | - | -928.80±10.463 | -393.68±3.9053 | **-340.36±3.9730** |
| | Time per step | 5.1879 | 3.8932 | 2.3587 | 2.7000 | 3.5935 |
| 100 | Filter RMSE | 0.1571±0.0017 | 0.2493±0.0122 | 0.1239±0.0006 | 0.1070±0.0001 | **0.1068±0.0001** |
| | ELBO (nats) | - | - | -4247.9±20.905 | -2069.7±11.814 | **-1794.7±5.4173** |
| | Time per step | 6.4546 | 4.6184 | 3.2697 | 4.5539 | 5.9263 |

(a) Before training

(b) After training

Figure 3: Frames predicted by rolling out $\text{NN}_\theta^f$ from two different starting points, before and after training. Between each frame, 3 transition steps are taken. Before training, no meaningful change is predicted but after training $\text{NN}_\theta^f$ predicts plausible movements.

shown in Figure 3. Before training, $\text{NN}_\theta^f$ predicts no meaningful change but after training it predicts movements the agent could realistically take. We quantify this further in Appendix B.3 by showing that the approximate average log likelihood $\ell_t(\theta)/t$ computed using Monte Carlo increases through training, thereby confirming our method can successfully learn high-dimensional model parameters of the agent movement in a fully online fashion.

## 6  Discussion

We have presented a novel online approach for variational state estimation and parameter learning. In our experiments, this methodology outperforms standard filtering approaches for high dimensional SSMs and is also able to learn online high dimensional model parameters of a neural network in a sequential VAE. However, it is not without its limitations. As with any stochastic variational inference technique, we can only obtain an accurate posterior approximation if the variational family is expressive enough and our stochastic gradient method finds a reasonable minimum. We also need to use flexible function approximators to keep the bias in our gradient estimates small. Finally, although our method is online, it can be quite computationally expensive in absolute terms as it requires solving an optimization problem and solving a regression task for each time step.

To reduce this computational cost, one can amortize the optimization cost for $\phi$ by learning a network taking a representation of observations up to now, $y^t$, and outputting $q_t^\phi$ as illustrated in Appendix D. Further work will investigate ways to also amortize the cost of function regression across time through a meta-learning strategy. From a theoretical point of view, it would be useful to establish conditions under which the proposed variational filter is exponentially stable and to study the asymptotic properties of the parameter learning procedure.

## Acknowledgments and Disclosure of Funding

The authors are grateful to Adrien Corenflos, Desi Ivanova and James Thornton for their comments. Andrew Campbell acknowledges support from the EPSRC CDT in Modern Statistics and Statistical Machine Learning (EP/S023151/1). Arnaud Doucet is partly supported by the EPSRC grant EP/R034710/1. He also acknowledges support of the UK Defence Science and Technology Laboratory (DSTL) and EPSRC under grant EP/R013616/1. This is part of the collaboration between US DOD, UK MOD and UK EPSRC under the Multidisciplinary University Research Initiative.

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
