# Appendix

## Table of Contents

The appendix is organized as follows. In Section A we cover in more detail some subtleties of our proposed methodology. In A.1, we motivate our choice of objectives for fitting our recursive gradient estimators, A.2 gives a full description of our algorithm and A.3 details the link between the recursions used in RL and those presented here. Section B gives experimental details as well as some further results for our Linear Gaussian (B.1), Chaotic Recurrent Neural Network (B.2) and Sequential VAE (B.3) experiments. The proofs for all our propositions are given in Section C. In Section D, we discuss methods for amortizing the cost of learning $\phi_t$ over time. We present possible architecture choices in D.1, alternative objectives in D.2, D.3, and notes on gradient computations in D.4. Finally, we discuss the broader impact of our research in Section E.

# A Methodological Details

## A.1 Objective for Recursive Fitting

Before specializing to our application, we first state a standard regression result. The following regression problem

$$\min_{h} \quad \mathbb{E}_{\rho(x,y)}\left[\|h(x) - k(y,x)\|_2^2\right] \tag{16}$$

has solution $h(x) = \mathbb{E}_{\rho(y|x)}[k(y,x)]$. Therefore, we can estimate $\mathbb{E}_{\rho(y|x)}[k(y,x)]$ by minimizing the following empirical version of the $L^2$ loss (16) (or a regularized version of it)

$$\min_{h \in \mathcal{F}} \quad \frac{1}{N} \sum_{i=1}^{N} \|h(x^i) - k(y^i, x^i)\|_2^2, \quad \text{where } x^i, y^i \overset{\text{i.i.d.}}{\sim} \rho(x,y)$$

over a flexible function class $\mathcal{F}$.

We note that if we make the following substitutions, then this solution exactly corresponds to the form of recursion (9) with our approximation $\hat{S}_{t+1}$ substituted for $S_{t+1}^{\theta,\phi_{1:t+1}}$:

$$x = x_{t+1}, \quad \rho(x) = q_{t+1}^{\phi_{t+1}}(x_{t+1}), \quad y = x_t, \quad \rho(y|x) = q_{t+1}^{\phi_{t+1}}(x_t|x_{t+1}),$$
$$h(x) = \hat{S}_{t+1}(x_{t+1}), \quad k(y,x) = \hat{S}_t(x_t) + s_{t+1}^{\theta}(x_t, x_{t+1}).$$

Similarly for fitting $\hat{T}_{t+1}$ we can make the following substitutions such that the regression solution corresponds to (10).

$$x = x_{t+1}, \quad \rho(x) = q_{t+1}^{\phi_{t+1}}(x_{t+1}), \quad y = \epsilon_t, \quad \rho(y|x) = \lambda(\epsilon_t), \quad h(x) = \hat{T}_{t+1}(x_{t+1}),$$
$$k(y,x) = \hat{T}_t(x_t(\phi_{t+1};\epsilon_t, x_{t+1}))\frac{\partial x_t(\phi_{t+1};\epsilon_t, x_{t+1})}{\partial x_{t+1}} + \nabla_{x_{t+1}} r_{t+1}^{\theta,\phi_{t:t+1}}(x_t(\phi_{t+1};\epsilon_t, x_{t+1}), x_{t+1}).$$

We note that the choice of distribution $\rho(x)$ is technically arbitrary, however, in practice, it will decide the region of space that our function approximation is most accurate. Therefore, we would like it to be close to the 'test time' distribution - the distribution of points that the approximators are evaluated at during the next time step. For $\hat{T}_{t+1}$, the test time distribution depends on $\phi_{t+2}$ which changes during optimization. This gives a series of test time distributions given by

$$\int q_{t+2}^{\phi_{t+2}}(x_{t+2}) q_{t+2}^{\phi_{t+2}}(x_{t+1}|x_{t+2}) dx_{t+2}. \tag{17}$$

Assuming an accurate variational approximation, this will approach the following one-step smoothing distribution: $p_\theta(x_{t+1}|y^{t+2})$. Our best approximation to this available at time $t+1$ is $q_{t+1}^{\phi_{t+1}}(x_{t+1})$ which approximates $p_\theta(x_{t+1}|y^{t+1})$. $\hat{S}_{t+1}$ is evaluated at the end of $\phi_{t+2}$ optimization so it has only two test time distributions, (17) with the final value of $\phi_{t+2}$ and $q_{t+1}^{\phi_{t+1}}(x_{t+1})$. Therefore, for both $\hat{T}_{t+1}$ and $\hat{S}_{t+1}$, we set $\rho(x) = q_{t+1}^{\phi_{t+1}}(x_{t+1})$. We found this to perform well in our experiments and so strategies to mitigate the distribution shift between training and test time were not needed. However, if necessary, we could artificially inflate the entropy of the training distribution to cover a wider region of the input space.

## A.2 Algorithm

The full detailed description of Algorithm 1 is shown in Algorithm 2.

Here, we have assumed the variational family is Gaussian and so we detail the reparameterization trick. We parameterize the standard deviation through $\log \sigma_t$ to ensure $\sigma_t > 0$.

We present both options for regression, neural networks and KRR. The exposition for KRR is as in [26]. When we use neural networks, we use $\varpi_t$ to represent the parameters of the $\hat{T}_t$ network and $\psi_t$ to represent the parameters of the $\hat{S}_t$ network. We describe L2 regression for the neural networks in Algorithm 2 although other losses are possible. KRR is based on a regularized L2 loss with regularization parameter $\lambda > 0$. It requires a kernel $k(x_t^i, x_t^j)$ that takes two input vectors and outputs a scalar similarity between them. In our experiments we use the Matérn kernel.

**Algorithm 2:** Online Variational Filtering and Parameter Learning - Full Algorithm Description

1  **for** $t = 1, \ldots, T$ **do**
2     Initialize $\phi_t$ e.g. $\phi_t \leftarrow \phi_{t-1}$
     /* Update $\phi_t$ using $M$ stochastic gradient steps         */
3     **for** $m = 1, \ldots, M$ **do**
        /* Split out variational parameters         */
4         $[\mu_t, \log \sigma_t, \tilde{\phi}_t] \leftarrow \phi_t$
        /* Sample $x_{t-1}$ and $x_t$ using the reparameterization trick    */
5         Sample $\epsilon_t^i \sim \mathcal{N}(0, \mathbb{I}_{d_x})$   for $i = 1, \ldots, N$
6         $x_t^i = \mu_t + \exp(\log \sigma_t)\epsilon_t^i$   for $i = 1, \ldots, N$    /* Element wise    */
7         $[\tilde{\mu}_t^i, \log \tilde{\sigma}_t^i] = Q^{\tilde{\phi}_t}(x_t^i)$   for $i = 1, \ldots, N$
        /* $Q^{\tilde{\phi}_t}$ is a function giving $q_t^{\phi_t}(x_{t-1}|x_t)$ statistics e.g. MLP    */
8         Sample $\tilde{\epsilon}_t^i \sim \mathcal{N}(0, \mathbb{I}_{d_x})$   for $i = 1, \ldots, N$
9         $x_{t-1}^i = \tilde{\mu}_t + \exp(\log \tilde{\sigma}_t)\tilde{\epsilon}_t^i$   for $i = 1, \ldots, N$   /* Element wise   */
10        $\phi_t \leftarrow \phi_t + \gamma_m \frac{1}{N} \sum_{i=1}^{N} \{\hat{T}_{t-1}(x_{t-1}^i)\frac{dx_{t-1}^i}{d\phi_t} + \nabla_{\phi_t} r_t(x_{t-1}^i, x_t^i)\}$
11     **end**
12

    /* Update $\hat{T}_t(x_t)$ and $\hat{S}_t(x_t)$ as in Section **3.5**       */
    /* Generate training datasets       */
13     Sample $x_{t-1}^i, x_t^i \sim q_t^{\phi_t}(x_{t-1}, x_t)$ as lines [4] to [9]   for $i = 1, \ldots, P$
14     $\mathcal{D}_T = \{x_t^i, \text{T-target}^i\}_{i=1}^{P}$ with T-target$^i = \hat{T}_{t-1}(x_{t-1}^i)\frac{\partial x_{t-1}^i}{\partial x_t^i} + \nabla_{x_t} r_t(x_{t-1}^i, x_t^i)$
15     $\mathcal{D}_S = \{x_t^i, \text{S-target}^i\}_{i=1}^{P}$ with S-target$^i = \hat{S}_{t-1}(x_{t-1}^i) + s_t^{\theta_{t-1}}(x_{t-1}^i, x_t^i)$
    /* Update function approximators       */
16     **if** *Regression using Neural Nets* **then**
17         **for** $j = 1, \ldots, J$ **do**
18             $\mathcal{I} \leftarrow$ minibatch sample from $\{1, \ldots, P\}$
19             $\varpi_t \leftarrow \varpi_t + \gamma_j \nabla_{\varpi_t} \frac{1}{|\mathcal{I}|} \sum_{i \in \mathcal{I}} \|\hat{T}_t^{\varpi_t}(x_t^i) - \text{T-target}^i\|_2^2$
20             $\psi_t \leftarrow \psi_t + \gamma_j \nabla_{\psi_t} \frac{1}{|\mathcal{I}|} \sum_{i \in \mathcal{I}} \|\hat{S}_t^{\psi_t}(x_t^i) - \text{S-target}^i\|_2^2$
21         **end**
22     **else if** *Regression using KRR* **then**
23         Let $\hat{T}_t(x^*) = R_{\mathcal{D}_T}(K_{\mathcal{D}_T} + P\lambda \mathbb{I}_P)^{-1} k_{\mathcal{D}_T}^*$
24         Let $\hat{S}_t(x^*) = R_{\mathcal{D}_S}(K_{\mathcal{D}_S} + P\lambda \mathbb{I}_P)^{-1} k_{\mathcal{D}_S}^*$
25         with
26           $R_{\mathcal{D}} = [\text{target}^1, \ldots, \text{target}^P] \in \mathbb{R}^{d_x \times P}$
27           $K_{\mathcal{D}} \in \mathbb{R}^{P \times P}, \ (K_{\mathcal{D}})_{ij} = k(x_t^i, x_t^j)$
28           $k_{\mathcal{D}}^* \in \mathbb{R}^{P}, \ (k_{\mathcal{D}}^*)_i = k(x_t^i, x_t^*)$
29     **end**
30

    /* Update $\theta$ using a stochastic gradient step       */
31     Sample $x_{t-1}^i, x_t^i \sim q_t^{\phi_t}(x_{t-1}, x_t), \ \ \tilde{x}_{t-1}^i \sim q_{t-1}^{\phi_{t-1}}(x_{t-1})$ for $i = 1, \ldots, N$
32     $\theta_t \leftarrow \theta_{t-1} + \eta_t \frac{1}{N} \sum_{i=1}^{N} \{\hat{S}_{t-1}(x_{t-1}^i) + s_t^{\theta_{t-1}}(x_{t-1}^i, x_t^i) - \hat{S}_{t-1}(\tilde{x}_{t-1}^i)\}$
33  **end**

---

## A.3   Link with Reinforcement Learning

The forward recursions in our method bear some similarity to the Bellman recursion present in RL. This is due to both relying on dynamic programming. We make explicit the relationship between our gradient recursions and the Bellman recursion in this section.

We first detail the standard RL framework and its Bellman recursion. The total expected reward we would like to optimize is

$$J(\phi) = \mathbb{E}_{\tau \sim p_\phi} \left[ \sum_{t=1}^{T} r(s_t, a_t) \right],$$

where $s_t$ is the state at time $t$, $a_t$ is the action taken at time $t$ and $r(s_t, a_t)$ is the reward for being in state $s_t$ and taking action $a_t$. This expectation is taken with respect to the trajectory distribution which is dependent on the policy $\pi_\phi$

$$p_\phi(\tau) = P(s_1)\pi_\phi(a_1|s_1) \prod_{t=2}^{T} P(s_t|s_{t-1}, a_{t-1})\pi_\phi(a_t|s_t).$$

The value function is then defined as the expected sum of future rewards when starting in state $s_t$ under policy $\pi_\phi$

$$V_t^{\text{RL}}(s_t) := \mathbb{E}\left[\sum_{k=t}^{T} r(s_k, a_k)\right].$$

This value function satisfies the following Bellman recursion

$$V_t^{\text{RL}}(s_t) = \mathbb{E}_{s_{t+1}, a_t \sim P(s_{t+1}|s_t, a_t)\pi_\phi(a_t|s_t)}\left[r(s_t, a_t) + V_{t+1}^{\text{RL}}(s_{t+1})\right].$$

The total expected reward $J(\phi)$ is then just the expected value of $V_1^{\text{RL}}$ taken with respect to the first state distribution

$$J(\phi) = \mathbb{E}_{s_1 \sim P(s_1)}\left[V_1^{\text{RL}}(s_1)\right].$$

For our application, we would like to instead have a *forward* recursion. A natural forward recursion appears when we consider an *anti-causal* graphical model for RL, where $s_t$ depends on $s_{t+1}$ and $a_{t+1}$; i.e. we consider the following *reverse*-time decomposition of the trajectory distribution

$$p_\phi(\tau) = P(s_T)\pi_\phi(a_T|s_T) \prod_{t=T-1}^{1} P(s_t|s_{t+1}, a_{t+1})\pi_\phi(a_t|s_t).$$

We define a new value function which is the sum of previous rewards

$$V_t(s_t) := \mathbb{E}\left[\sum_{k=1}^{t} r(s_k, a_k)\right].$$

It follows a corresponding forward Bellman-type recursion

$$V_{t+1}(s_{t+1}) = \mathbb{E}_{\pi_\phi(a_{t+1}|s_{t+1})P(s_t|s_{t+1}, a_{t+1})}\left[r(s_{t+1}, a_{t+1}) + V_t(s_t)\right].$$

$J(\phi)$ is then now the expected value of $V_T$ taken with respect to the final state distribution

$$J(\phi) = \mathbb{E}_{s_T \sim P(s_T)}\left[V_T(s_T)\right].$$

This forward Bellman recursion is non-standard in the literature but is useful when we adapt it for our application. We define $s_t = x_t$, $a_t = x_{t-1} \sim q_t^\phi(x_{t-1}|x_t)$ and $P(s_t|s_{t+1}, a_{t+1}) = \delta(s_t = a_{t+1})$. The 'reward' is defined as

$$r(s_t, a_t) = r(x_t, x_{t-1}) = \log \frac{f(x_t|x_{t-1})g(y_t|x_t)q_{t-1}^{\phi_{t-1}}(x_{t-1})}{q_t^{\phi_t}(x_t)q_t^{\phi_t}(x_{t-1}|x_t)},$$

$$r(s_1, a_1) = r(x_1, x_0) = \log \frac{\mu(x_1)g(y_1|x_1)}{q_1^{\phi_1}(x_1)},$$

where we have suppressed $\theta$ from the notation for conciseness. Note $a_1 = x_0$ has no meaning here. The 'policy' is defined as the backward kernel

$$\pi_\phi(a_t|s_t) = q_t^{\phi_t}(x_{t-1}|x_t).$$

With these definitions, the trajectory distribution is

$$p_\phi(\tau) = q_T^{\phi_T}(x_T) \prod_{t=T}^{1} q_t^{\phi_t}(x_{t-1}|x_t).$$

(Since $x_0$ has no meaning in our application, the final $q_1^{\phi_1}(x_0|x_1)$ distribution that appears has no significance.) With this formulation, the sum of rewards now corresponds to the ELBO which we would like to maximize with respect to $\phi$

$$\mathcal{L}_T = \mathbb{E}_{p_\phi(\tau)} \left[ \sum_{t=1}^T r(s_t, a_t) \right].$$

Just as in our anti-causal RL example, this can be broken down into a value function that summarizes previous rewards

$$V_{t+1}(x_{t+1}) = \mathbb{E}\left[ \sum_{k=1}^{t+1} r(s_k, a_k) \right] = \mathbb{E}_{q_{t+1}^{\phi_{1:t+1}}(x_{1:t}|x_{t+1})} \left[ \log \frac{p_\theta(x_{1:t+1}, y^{t+1})}{q_{t+1}^{\phi_{1:t+1}}(x_{1:t+1})} \right],$$

$$\mathcal{L}_T = \mathbb{E}_{q_T^{\phi_T}(x_T)} \left[ V_T(x_T) \right].$$

This follows a forward Bellman recursion (equation (7) in the main text).

$$V_{t+1}(x_{t+1}) = \mathbb{E}_{q_{t+1}^{\phi_{t+1}}(x_t|x_{t+1})} \left[ \log \frac{f(x_{t+1}|x_t)g(y_{t+1}|x_{t+1})q_t^{\phi_t}(x_t)}{q_{t+1}^{\phi_{t+1}}(x_{t+1})q_{t+1}^{\phi_{t+1}}(x_t|x_{t+1})} + V_t(x_t) \right].$$

Since we would like to optimize the ELBO rather than just evaluate it, we do not make use of $V_t(x_t)$ directly. We instead differentiate this forward in time Bellman recursion to obtain our gradient recursions. To obtain equation (9) in the paper we differentiate with respect to $\theta$. To obtain equation (10) we differentiate with respect to $x_t$, we then use $\frac{\partial}{\partial x_t} V_t(x_t)$ to get an equation for $\nabla_{\phi_{t+1}} V_{t+1}(x_{t+1})$.

Our approach here is complementary to that of [14, 25] but differs in the fact we focus on forward in time recursions allowing an online optimization of the ELBO. [25] and subsequent work focus on fitting RL into a probabilistic context whereas we take ideas from RL (recursive function estimation) to enable online inference. We note that [43] also define suitable rewards to fit probabilistic inference into an RL framework but again only focus on backward Bellman recursions.

## B    Experiment Details

### B.1    Linear Gaussian Experiment

For both experiments, we randomly initialize $F$ and $G$ to have eigenvalues between 0.5 and 1.0. When learning $\phi$ we set the diagonals of $U$ and $V$ to be $0.1^2$. We use a learning rate of 0.01 over 5000 iterations for each time step. We decay the learning rate by 0.999 at every inner training step. For the initial time point we use a learning rate of 0.1 with the same number of iterations and decay because the $\phi$ parameters start far away from the optimum but for the proceeding time points we initialize at the previous solution hence they are already close to the local optimum. We represent $\hat{T}_t$ using KRR, and use 500 data points to perform the fitting at each time step. We set the regularization parameter to 0.1. We use an RBF kernel with a bandwidth learned by minimizing mean squared error on an additional validation dataset.

For learning $\phi$ and $\theta$ jointly, we set the diagonals of $V$ to be $0.25^2$. We compare different training methods against the RMLE baseline under the same settings. To learn $\phi$, we use a learning rate of 0.01 over 500 iterations for each time step with a learning rate decay of 0.991. We represent $\hat{T}_t$ using KRR, and use 512 data points to perform the fitting at each time step. We set the regularization parameter to 0.01. To learn $\theta$, we use a learning rate of 1e-2 and a Robbins-Monro type learning rate decay. We represent $\hat{S}_t$ using KRR with 1000 data points and regularization parameter 1e-4. The experiments were run on an internal CPU cluster with Intel Xeon E5-2690 v4 CPU.

### B.2    Chaotic Recurrent Neural Network

We follow closely [44] using the same parameter settings for data generation. We use 1 million particles for EnKF and BPF for dimension $d_x = 5$ and 20, and 250000 particles for dimension

Table 2: Root Mean Squared Error between the 'ground truth' posterior mean and variational mean estimates for the filtering distribution $p(x_t|y^t)$ and one-step smoothing distribution $p(x_{t-1}|y^t)$ over 10 runs.

| $d_x$ | | AELBO-1 | AELBO-2 | Ours |
|---|---|---|---|---|
| 5 | Filter mean RMSE | 0.0644±0.0037 | 0.0155±0.0006 | **0.0128±0.0007** |
| | 1-step mean RMSE | - | 0.0241±0.0008 | **0.0202±0.0009** |

$d_x = 100$ in order to match the computation cost. For variational methods, we train for 500 iterations at each timestep with minibatch size 10 and learning rate 1e-2, and we use a single-layer MLP with 100 neurons to parameterize each $q_t^{\phi_t}(x_{t-1}|x_t)$ for AELBO-2 and our method. The function $\hat{T}_t$ is represented using KRR with 100 samples for $d_x = 5$ and 250 samples for $d_x = 20$ and 100. The regularization strength is fixed to be 0.1 while the kernel bandwidth is learned at each timestep on a validation dataset for 25 iterations with minibatch size 10 and learning rate 1e-2. The extra time for optimizing the kernel parameter is included in the presented runtime results. The experiments were run on an internal CPU cluster with Intel Xeon E5-2690 v4 CPU.

For $d_x = 5$, we further confirm the gain in ELBO by comparing the variational means of $q_t^{\phi_t}(x_t)$ and $q_t^{\phi_t}(x_{t-1}) = \int q_t^{\phi_t}(x_t)q_t^{\phi_t}(x_{t-1}|x_t)\mathrm{d}x_t$ against the 'ground truth' posterior means of $p(x_t|y^t)$ and $p(x_{t-1}|y^t)$ computed using PF with 10 million particles. As shown in Table 2, our method attains a significantly lower error in terms of both metrics.

For our comparison with the offline ELBO, we verify our conclusion holds by considering a range of seeds and examining all dimensions of the variational joint. We plot the two joint variational distributions alongside the true hidden state in Figure 4 for 3 seeds and 5 dimensions. We see that in all cases the joint variational distributions trained with our online ELBO and the offline ELBO match very closely.

We also note that for longer time intervals, using the offline ELBO objective results in training instabilities due to the necessity of rolling out the entire backward joint variational distribution. To train for 100 time steps, a 'warm start' was needed whereby we first train for a small number of training iterations on each of a series of intermediary objectives. If the total number of time steps is $t$, then there are $t$ intermediary objectives each being an offline ELBO with $\tau$ terms for $\tau = 1, 2, \ldots, t$. After the warm start, many gradient updates are taken using the offline ELBO corresponding to the full $t$ steps.

## B.3   Sequential Variational Auto-Encoder Experiment

A video demonstration of the sequential VAE model is available at https://github.com/andrew-cr/online_var_fil.

### B.3.1   Experimental Details

We parameterize $\mathrm{NN}_\theta^f$ as a residual MLP which consists of 4 stacked layers of the form $f(x) = x + s\mathrm{MLP}(x)$ where $\mathrm{MLP}(x)$ is an MLP with a single hidden layer of hidden dimension 32 and $s$ is a learned scaling parameter. We parameterize $\mathrm{NN}^g$ as a convolutional neural network, using the architecture suggested by [5] and implemented in PyTorch in [6]. Specifically, the architecture consists of the following layers:

- Convolutional layer, 128 outputs channels, kernel size of 3 and padding of 1
- ReLU activation
- Two residual blocks consisting of

---

[5] https://github.com/deepmind/sonnet/blob/v2/examples/vqvae_example.ipynb
[6] https://github.com/karpathy/deep-vector-quantization MIT License

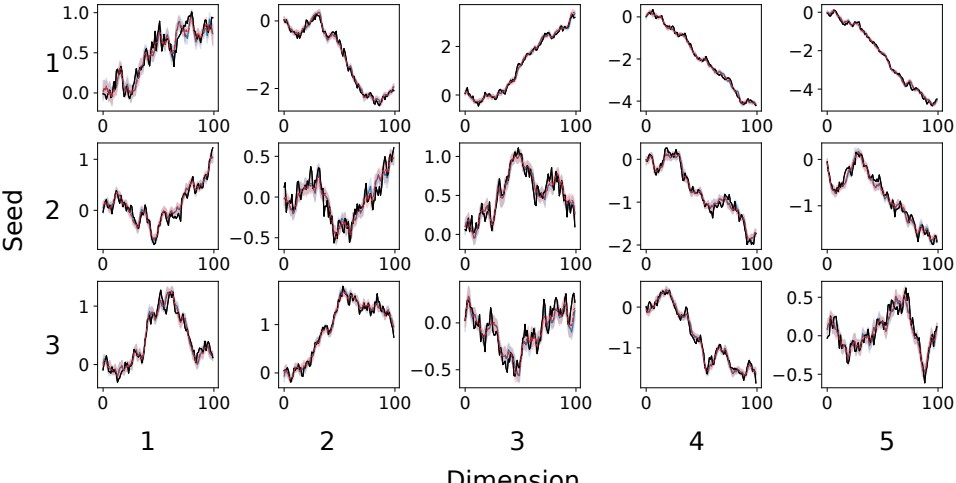

Figure 4: Comparison between joint variational distributions trained offline (red dashed) versus online (blue full). The data is generated as in the Chaotic Recurrent Neural Network example, for 10 time steps and is shown in black. Colored lines show variational means through time whilst the shaded regions represent $\pm 1$ std. The row of a plot corresponds to the seed used to generate the data whilst the column corresponds to the dimension in this 5 dimensional tracking example.

- – Convolutional layer, 32 outputs channels, kernel size of 3, padding of 1
- – ReLU activation
- – Convolutional layer, 128 output channels, kernel size of 1, padding of 0
- – Residual connection
- • Transposed convolution, 64 output channels, kernel size of 4, stride of 2 and padding of 1
- • ReLU activation
- • Transposed convolution, 3 output channels, kernel size of 4, stride of 2 and padding of 1

To pre-train the decoder we use the encoder architecture from the same sources and train using the standard minibatch ELBO objective to convergence.

We use KRR to represent $\hat{T}_t$. We use a KRR regularization parameter of $0.1$ and use an RBF kernel with learned bandwidth on a validation dataset. We update $\hat{T}_t$ using a dataset of size 512. For $\hat{S}_t$ we use a two hidden layer MLP with ReLU activations with hidden layer dimensions, 256 and 1024. The $\theta$ dimension is 8452. The MLP has an output dimension of 8453, the first 8452 outputs give the gradient direction and the last gives the log magnitude of the gradient. The MLP is trained on the regression dataset using a combination of a direction and magnitude loss. The direction loss is the negative cosine similarity whilst the magnitude loss is an MSE loss in log magnitude space. The two losses are then combined with equal weighting. This separation of the gradient into a magnitude and direction is similar to the gradient pre-processing described in [1]. The dataset size used for regression is 1024. We take minibatches of size 32 randomly sampled from this dataset and take 128 gradient steps on the $\hat{S}_t$ weights for each time step with a learning rate of 0.001.

We use mean field $q_t^{\phi_t}(x_t)$ and $q_t^{\phi_t}(x_{t-1}|x_t)$. The mean vector for $q_t^{\phi_t}(x_{t-1}|x_t)$ is given by an MLP with input $x_t$, and two hidden layers of dimension 64. The log standard deviation for $q_t^{\phi_t}(x_{t-1}|x_t)$ is learned directly and does not depend on $x_t$.

We set $U = V = 0.1\mathbb{I}$. We use a learning rate for $\theta$ updates of 0.001. We run 200 iterations of inner $\phi_t$ optimization at each time step, with a learning rate of 0.03 for $q_t^{\phi_t}(x_t)$ statistics and 0.003 for $q_t^{\phi_t}(x_{t-1}|x_t)$ weights. These experiments were run on a single RTX 3090 GPU.

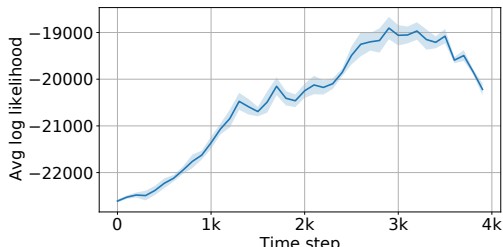

Figure 5: $\hat{\ell}_t^q(\theta_k)/t$ for $k = [1, \ldots, t]$. Solid line is mean over three different seeds using the same dataset, transparent area is $\pm$ one standard deviation.

### B.3.2  Average log likelihood

To quantify the quality of the learned transition function $\mathrm{NN}_\theta^f$ we can compute the approximate average log likelihood $\ell_t(\theta)/t$ using values of $\theta$ from different stages of training. The environment is highly non-stationary on the timescale considered because the dataset consists of video frames from an agent exploring a maze giving transitions that are diverse and not repeated. We therefore do not expect $\ell_t(\theta)/t$ to converge to a constant but its general trajectory is useful to quantify the agreement between the model and the observations. $\ell_t(\theta)/t$ is defined as

$$\frac{\ell_t(\theta)}{t} = \frac{1}{t}\sum_{k=1}^t \log p_\theta(y_k|y^{k-1}) = \frac{1}{t}\sum_{k=1}^t \log \int p_\theta(x_{k-1}|y^{k-1})f_\theta(x_k|x_{k-1})g(y_k|x_k)\mathrm{d}x_k\mathrm{d}x_{k-1}.$$

We approximate this quantity using the learned filtering distributions

$$\frac{\ell_t^q(\theta)}{t} = \frac{1}{t}\sum_{k=1}^t \log \int q_{k-1}^{\phi_{k-1}}(x_{k-1})f_\theta(x_k|x_{k-1})g(y_k|x_k)\mathrm{d}x_k\mathrm{d}x_{k-1}$$

which we can estimate through Monte Carlo

$$\frac{\hat{\ell}_t^q(\theta)}{t} = \frac{1}{t}\sum_{k=1}^t \log\left\{\sum_{i=1}^N g(y_k|x_k^n)\right\} \quad x_{k-1}^n, x_k^n \sim q_{k-1}^{\phi_{k-1}}(x_{k-1})f_\theta(x_k|x_{k-1}).$$

We note that $\hat{\ell}_t^q/t$ depends on $\phi_{1:t}$ so can only be computed at the end of the run. To monitor progress of $\theta$, we compute $\hat{\ell}_t^q(\theta_k)/t$ for $k = [1, \ldots, t]$ where $\theta_k$ is the value of the model parameters during training at time step $k$. We plot the results in Figure 5. We find that indeed $\hat{\ell}_t^q(\theta_k)/t$ generally increases through training showing that our method can learn high dimensional model parameters. We conjecture that the few decreases in $\hat{\ell}_t^q(\theta_k)/t$ are due to online nature of the algorithm, if the current temporally local state of the system involves transitions that are not represented throughout the training sequence then updates in this region of time will cause an overall decrease in log likelihood when aggregated over the entire sequence. However, in the limit as $t \to \infty$ a recursive maximum likelihood approach, upon which our method is based, is expected to reach a local maximum of $\ell_t(\theta)/t$ under regularity conditions because local transients are averaged out in the long term.

## C   Proofs

### C.1   Proof of Proposition 1

Recall from (6) that we have

$$q_{t+1}^{\phi}(x_{1:t+1}) = q_t^{\phi}(x_{1:t})m_{t+1}^{\phi}(x_{t+1}|x_t),$$

so

$$\log \frac{p_\theta(x_{1:t+1}, y^{t+1})}{q_{t+1}^{\phi}(x_{1:t+1})} = \log \frac{p_\theta(x_{1:t}, y^t)}{q_t^{\phi}(x_{1:t})} + \log \frac{f_\theta(x_{t+1}|x_t)g_\theta(y_{t+1}|x_{t+1})}{m_{t+1}^{\phi}(x_{t+1}|x_t)}$$

$$= \log \frac{p_\theta(x_{1:t}, y^t)}{q_t^{\phi}(x_{1:t})} + r_{t+1}^{\theta,\phi}(x_t, x_{t+1}).$$

From the definition

$$V_{t+1}^{\theta,\phi}(x_{t+1}) = \mathbb{E}_{q_{t+1}^{\phi}(x_{1:t}|x_{t+1})}\left[\log \frac{p_\theta(x_{1:t+1}, y^{t+1})}{q_{t+1}^{\phi}(x_{1:t+1})}\right],$$

we have directly

$$\mathcal{L}_{t+1}(\theta, \phi) = \mathbb{E}_{q_{t+1}^{\phi}(x_{t+1})}\left[V_{t+1}^{\theta,\phi}(x_{t+1})\right]$$

and, crucially,

$$V_{t+1}^{\theta,\phi}(x_{t+1}) := \mathbb{E}_{q_{t+1}^{\phi}(x_{1:t}|x_{t+1})}\left[\log \frac{p_\theta(x_{1:t}, y^t)}{q_t^{\phi}(x_{1:t})} + r_{t+1}^{\theta,\phi}(x_t, x_{t+1})\right]$$

$$= \mathbb{E}_{q_{t+1}^{\phi}(x_t|x_{t+1})}\left[\mathbb{E}_{q_t^{\phi}(x_{1:t-1}|x_t)}\left[\log \frac{p_\theta(x_{1:t}, y^t)}{q_t^{\phi}(x_{1:t})}\right] + r_{t+1}^{\theta,\phi}(x_t, x_{t+1})\right]$$

$$= \mathbb{E}_{q_{t+1}^{\phi}(x_t|x_{t+1})}\left[V_t^{\theta,\phi}(x_t) + r_{t+1}^{\theta,\phi}(x_t, x_{t+1})\right].$$

### C.2   Proof of Proposition 2

We have by direct differentiation that

$$\nabla_\theta \mathcal{L}_t(\theta, \phi) = \mathbb{E}_{q_t^{\phi}(x_t)}[\nabla_\theta V_t^{\theta,\phi}(x_t)]$$

$$= \mathbb{E}_{q_t^{\phi}(x_{1:t})}\left[\nabla_\theta \log p_\theta(x_{1:t}, y^t)\right]$$

$$= \mathbb{E}_{q_t^{\phi}(x_t)}\left[S_t^{\theta,\phi}(x_t)\right]$$

where

$$S_t^{\theta,\phi}(x_t) := \mathbb{E}_{q_t^{\phi}(x_{1:t-1}|x_t)}\left[\nabla_\theta \log p_\theta(x_{1:t}, y^t)\right].$$

This quantity satisfies the recursion

$$S_{t+1}^{\theta,\phi}(x_{t+1}) = \mathbb{E}_{q_{t+1}^{\phi}(x_{1:t}|x_{t+1})}\left[\nabla_\theta \log p_\theta(x_{1:t+1}, y^{t+1})\right]$$

$$= \mathbb{E}_{q_{t+1}^{\phi}(x_{1:t}|x_{t+1})}\left[\nabla_\theta \log p_\theta(x_{1:t}, y^t) + s_{t+1}^{\theta}(x_t, x_{t+1})\right]$$

$$= \mathbb{E}_{q_{t+1}^{\phi}(x_t|x_{t+1})}\left[\mathbb{E}_{q_t^{\phi}(x_{1:t-1}|x_t)}\left[\nabla_\theta \log p_\theta(x_{1:t}, y^t)\right] + s_{t+1}^{\theta}(x_t, x_{t+1})\right]$$

$$= \mathbb{E}_{q_{t+1}^{\phi}(x_t|x_{t+1})}\left[S_t^{\theta,\phi}(x_t) + s_{t+1}^{\theta}(x_t, x_{t+1})\right]$$

for

$$s_{t+1}^{\theta}(x_t, x_{t+1}) := \nabla_\theta \log f_\theta(x_{t+1}|x_t)g_\theta(y_{t+1}|x_{t+1}).$$

## C.3 Proof of Proposition 3

We have by a direct application of the reparameterization trick that

$$\nabla_{\phi_t}\mathcal{L}_t(\theta,\phi_{1:t}) = \mathbb{E}_{\lambda(\epsilon_t)}[\nabla_{\phi_t}V_t^{\theta,\phi_{1:t}}(x_t(\phi_t;\epsilon_t))],$$

where $V_t^{\theta,\phi_{1:t}}(x_t) := \mathbb{E}_{q_t^{\phi_{1:t}}(x_{1:t-1}|x_t)}\left[\log\frac{p_\theta(x_{1:t},y^t)}{q_t^{\phi_{1:t}}(x_{1:t})}\right]$. We have

$$V_{t+1}^{\theta,\phi_{1:t+1}}(x_{t+1}) = \mathbb{E}_{q_{t+1}^{\phi_{t+1}}(x_t|x_{t+1})}\left[V_t^{\theta,\phi_{1:t}}(x_t) + r_{t+1}^{\theta,\phi_{t:t+1}}(x_t,x_{t+1})\right]$$

$$= \mathbb{E}_{\lambda(\epsilon_t)}\left[V_t^{\theta,\phi_{1:t}}(x_t(\phi_{t+1};\epsilon_t,x_{t+1})) + r_{t+1}^{\theta,\phi_{t:t+1}}(x_t(\phi_{t+1};\epsilon_t,x_{t+1}),x_{t+1})\right]$$

Hence,

$$\nabla_{\phi_{t+1}}V_{t+1}^{\theta,\phi_{1:t+1}}(x_{t+1}(\phi_{t+1};\epsilon_{t+1}))$$

$$= \mathbb{E}_{\lambda(\epsilon_t)}\left[\left.\frac{\partial}{\partial x_t}V_t^{\theta,\phi_{1:t}}(x_t)\right|_{x_t=x_t(\phi_{t+1};\epsilon_t,x_{t+1}(\phi_{t+1};\epsilon_{t+1}))}\frac{\mathrm{d}x_t(\phi_{t+1};\epsilon_t,x_{t+1}(\phi_{t+1};\epsilon_{t+1}))}{\mathrm{d}\phi_{t+1}}\right.$$

$$\left.+ \nabla_{\phi_{t+1}}r_{t+1}^{\theta,\phi_{t:t+1}}(x_t(\phi_{t+1};\epsilon_t,x_{t+1}(\phi_{t+1};\epsilon_{t+1})),x_{t+1}(\phi_{t+1};\epsilon_{t+1}))\right],$$

where

$$\left.\frac{\partial}{\partial x_t}V_t^{\theta,\phi_{1:t}}(x_t)\right|_{x_t=x_t(\phi_{t+1};\epsilon_t,x_{t+1}(\phi_{t+1};\epsilon_{t+1}))} = T_t^{\theta,\phi_{1:t}}(x_t(\phi_{t+1};\epsilon_t,x_{t+1}(\phi_{t+1};\epsilon_{t+1}))).$$

For the forward recursion,

$$T_{t+1}^{\theta,\phi_{1:t+1}}(x_{t+1}) = \frac{\partial}{\partial x_{t+1}}V_{t+1}^{\theta,\phi_{1:t+1}}(x_{t+1})$$

$$= \mathbb{E}_{\lambda(\epsilon_t)}\left[\frac{\partial}{\partial x_{t+1}}V_t^{\theta,\phi_{1:t}}(x_t(\phi_{t+1};\epsilon_t,x_{t+1})) + \nabla_{x_{t+1}}r_{t+1}^{\theta,\phi_{t:t+1}}(x_t(\phi_{t+1};\epsilon_t,x_{t+1}),x_{t+1})\right]$$

$$= \mathbb{E}_{\lambda(\epsilon_t)}\left[\left.\frac{\partial}{\partial x_t}V_t^{\theta,\phi_{1:t}}(x_t)\right|_{x_t=x_t(\phi_{t+1};\epsilon_t,x_{t+1})}\frac{\partial x_t(\phi_{t+1};\epsilon_t,x_{t+1})}{\partial x_{t+1}}\right.$$

$$\left.+ \nabla_{x_{t+1}}r_{t+1}^{\theta,\phi_{t:t+1}}(x_t(\phi_{t+1};\epsilon_t,x_{t+1}),x_{t+1})\right],$$

where again

$$\left.\frac{\partial}{\partial x_t}V_t^{\theta,\phi_{1:t}}(x_t)\right|_{x_t=x_t(\phi_{t+1};\epsilon_t,x_{t+1})} = T_t^{\theta,\phi_{1:t}}(x_t(\phi_{t+1};\epsilon_t,x_{t+1})).$$

Here, $\frac{\mathrm{d}x_t(\phi_{t+1};\epsilon_t,x_{t+1}(\phi_{t+1};\epsilon_{t+1}))}{\mathrm{d}\phi_{t+1}}$, $\nabla_{\phi_{t+1}}r_{t+1}^{\theta,\phi_{t:t+1}}(x_t(\phi_{t+1};\epsilon_t,x_{t+1}(\phi_{t+1};\epsilon_{t+1})),x_{t+1}(\phi_{t+1};\epsilon_{t+1}))$, $\nabla_{x_{t+1}}r_{t+1}^{\theta,\phi_{t:t+1}}(x_t(\phi_{t+1};\epsilon_t,x_{t+1}),x_{t+1})$ denote total derivatives w.r.t. $\phi_{t+1}$ and $x_{t+1}$, which can all be computed using the reparameterization trick.

# D  Amortization

In the main paper, we have focused on the case where we use a new set of variational parameters $\phi_t$ at each time step. This is conceptually simple and easy to use; however, it requires a new inner optimization run for each time step. In this section, we describe methods to amortize the $\phi$ optimization over time. This entails learning an amortization network which takes in $y$-observations and gives variational distribution statistics. With this network in hand, information from previous time steps regarding the relation between the $y$-observations and the variational statistics can be re-used. This results in computational savings over a method which treats each time step in isolation.

## D.1  Architecture

The variational distributions of interest $q_t^\phi(x_t|y^t)$ and $q_t^\phi(x_{t-1}|y^{t-1}, x_t)$ approximate $p_\theta(x_t|y^t)$ and $p_\theta(x_{t-1}|y^{t-1}, x_t)$ respectively. Therefore, to produce the $q_t^\phi$ statistics, we use a Recurrent Neural Network (RNN) to first encode the sequence of observations, $y^t$, creating a representation $h_t$. Then a network takes $h_t$ to give $q_t^\phi(x_t|y^t)$ statistics and a separate network takes $h_{t-1}$ and $x_t$ to give $q_t^\phi(x_{t-1}|y^{t-1}, x_t)$ statistics.

In the simplest version of this architecture, we can directly take $h_t$ to be the statistics of the filtering distribution $q_t^\phi(x_t|y^t)$. In this case, two amortization networks are learned: one forward filtering network learns to output the filtering statistics $h_t$ given $h_{t-1}$ and $y_t$; another backward smoothing network learns to approximate the backward kernel $p_\theta(x_{t-1}|y^{t-1}, x_t)$ given $h_{t-1}$ and $x_t$. Note that when we fix and detach $h_{t-1}$ at each time $t$, in a way this works like supervised learning, i.e. to learn to produce the same filtering and smoothing parameters as the non-amortized case, so the objective of the non-amortized case can be directly translated here to learn amortized networks.

## D.2  Amortized Joint ELBO Maximization Approach

One issue with using the non-amortized objective directly is that the amortized backward smoothing kernels $q_t^\phi(x_{t-1}|y^{t-1}, x_t)$ are not learned to jointly maximize the joint ELBO $\mathcal{L}_T(\theta, \phi)$ over time. Here, we give an alternative, more rigorous treatment of the amortization objective. We assume in this section that we take $h_t$ as the filtering statistics and detach $h_t$ at each time step.

For the amortized forward filtering network, we would like to maximize the objective $-\sum_{t=1}^T \text{KL}\left(q_t^\phi(x_t|y^t) \parallel p_\theta(x_t|y^t)\right)$. Since the KL is intractable, we learn $q_t^\phi(x_t|y^t)$ by maximizing $\mathcal{L}_t(\theta, \phi)$ at each time step $t$. This is the same as the non-amortized case, which only utilizes the functional approximator $T_t^{\theta,\phi}(x_t)$.

For the amortized backward smoothing network, we would like to maximize $\mathcal{L}_T(\theta, \phi)$ jointly over time. To do this, we can similarly optimize $\mathcal{L}_t(\theta, \phi) - \mathcal{L}_{t-1}(\theta, \phi)$ at each time $t$ similar to learning $\theta$. For the gradients $\nabla_\phi \mathcal{L}_t(\theta, \phi)$, we have the following result:

**Proposition 4.** *The ELBO gradient $\nabla_\phi \mathcal{L}_t(\theta, \phi)$ satisfies for $t \geq 1$*

$$\nabla_\phi \mathcal{L}_t(\theta, \phi) = \nabla_\phi \mathbb{E}_{q_t^\phi(x_t|y^t)}[V_t^{\theta,\phi}(x_t)] = \mathbb{E}_{\lambda(\epsilon_t)}[\nabla_\phi V_t^{\theta,\phi}(x_t(\phi; \epsilon_t))].$$

*Additionally, one has*

$$\nabla_\phi V_{t+1}^{\theta,\phi}(x_{t+1}(\phi; \epsilon_{t+1})) = \mathbb{E}_{\lambda(\epsilon_t)}\left[ T_t^{\theta,\phi}(x_t(\phi; \epsilon_t, x_{t+1}(\phi; \epsilon_{t+1}))) \frac{\mathrm{d}x_t(\phi; \epsilon_t, x_{t+1}(\phi; \epsilon_{t+1}))}{\mathrm{d}\phi} \right.$$
$$\left. + U_t^{\theta,\phi}(x_t(\phi; \epsilon_t, x_{t+1}(\phi; \epsilon_{t+1}))) + \nabla_\phi r_{t+1}^{\theta,\phi}(x_t(\phi; \epsilon_t, x_{t+1}(\phi; \epsilon_{t+1})), x_{t+1}(\phi; \epsilon_{t+1})) \right],$$

*where $T_t^{\theta,\phi}(x_t) := \frac{\partial}{\partial x_t} V_t^{\theta,\phi}(x_t)$, $U_t^{\theta,\phi}(x_t) := \nabla_\phi V_t^{\theta,\phi}(x_t)$ satisfy the forward recursions*

$$T_{t+1}^{\theta,\phi}(x_{t+1}) = \mathbb{E}_{\lambda(\epsilon_t)}\left[ T_t^{\theta,\phi}(x_t(\phi; \epsilon_t, x_{t+1})) \frac{\partial x_t(\phi; \epsilon_t, x_{t+1})}{\partial x_{t+1}} \right.$$
$$\left. + \nabla_{x_{t+1}} r_{t+1}^{\theta,\phi}(x_t(\phi; \epsilon_t, x_{t+1}), x_{t+1}) \right],$$

$$U_{t+1}^{\theta,\phi}(x_{t+1}) = \mathbb{E}_{\lambda(\epsilon_t)}\left[T_t^{\theta,\phi}(x_t(\phi;\epsilon_t,x_{t+1}))\frac{\partial x_t(\phi;\epsilon_t,x_{t+1})}{\partial\phi}\right.$$
$$\left. + U_t^{\theta,\phi}(x_t(\phi;\epsilon_t,x_{t+1})) + \nabla_\phi r_{t+1}^{\theta,\phi}(x_t(\phi;\epsilon_t,x_{t+1}),x_{t+1})\right].$$

*Proof.* By a direct application of the reparameterization trick,
$$\nabla_\phi\mathcal{L}_t(\theta,\phi) = \mathbb{E}_{\lambda(\epsilon_t)}[\nabla_\phi V_t^{\theta,\phi}(x_t(\phi;\epsilon_t))].$$
By Proposition 1,
$$V_{t+1}^{\theta,\phi}(x_{t+1}) = \mathbb{E}_{q_{t+1}^\phi(x_t|y^t,x_{t+1})}[V_t^{\theta,\phi}(x_t) + r_{t+1}^{\theta,\phi}(x_t,x_{t+1})].$$
Hence, using the chain rule,

$$\nabla_\phi V_{t+1}^{\theta,\phi}(x_{t+1}(\phi;\epsilon_{t+1})) = \mathbb{E}_{\lambda(\epsilon_t)}\left[\frac{\partial}{\partial x_t}V_t^{\theta,\phi}(x_t)\right|_{x_t=x_t(\phi;\epsilon_t,x_{t+1}(\phi;\epsilon_{t+1}))}\frac{dx_t(\phi;\epsilon_t,x_{t+1}(\phi;\epsilon_{t+1}))}{d\phi}$$
$$+ \frac{\partial}{\partial\phi}V_t^{\theta,\phi}(x_t)\Big|_{x_t=x_t(\phi;\epsilon_t,x_{t+1}(\phi;\epsilon_{t+1}))} + \nabla_\phi r_{t+1}^{\theta,\phi}(x_t(\phi;\epsilon_t,x_{t+1}(\phi;\epsilon_{t+1})),x_{t+1}(\phi;\epsilon_{t+1}))\Big].$$

The functions $\frac{\partial}{\partial x_t}V_t^{\theta,\phi}(x_t), \frac{\partial}{\partial\phi}V_t^{\theta,\phi}(x_t)$ are defined as $T_t^{\theta,\phi}(x_t), U_t^{\theta,\phi}(x_t)$ respectively.

$T_t^{\theta,\phi}(x_t)$ follows the same forward recursion as in Proposition 3. For $U_t^{\theta,\phi}(x_t)$, we have

$$U_{t+1}^{\theta,\phi}(x_{t+1}) = \frac{\partial}{\partial\phi}V_{t+1}^{\theta,\phi}(x_{t+1})$$
$$= \frac{\partial}{\partial\phi}\mathbb{E}_{q_{t+1}^\phi(x_t|y^t,x_{t+1})}[V_t^{\theta,\phi}(x_t) + r_{t+1}^{\theta,\phi}(x_t,x_{t+1})]$$
$$= \mathbb{E}_{\lambda(\epsilon_t)}\left[\nabla_\phi\left(V_t^{\theta,\phi}(x_t(\phi;\epsilon_t,x_{t+1})) + r_{t+1}^{\theta,\phi}(x_t(\phi;\epsilon_t,x_{t+1}),x_{t+1})\right)\right]$$
$$= \mathbb{E}_{\lambda(\epsilon_t)}\left[\frac{\partial}{\partial x_t}V_t^{\theta,\phi}(x_t)\Big|_{x_t=x_t(\phi;\epsilon_t,x_{t+1})}\frac{\partial x_t(\phi;\epsilon_t,x_{t+1})}{\partial\phi}\right.$$
$$\left.+ \frac{\partial}{\partial\phi}V_t^{\theta,\phi}(x_t)\Big|_{x_t=x_t(\phi;\epsilon_t,x_{t+1})} + \nabla_\phi r_{t+1}^{\theta,\phi}(x_t(\phi;\epsilon_t,x_{t+1}),x_{t+1})\right]$$
$$= \mathbb{E}_{\lambda(\epsilon_t)}\left[T_t^{\theta,\phi}(x_t(\phi;\epsilon_t,x_{t+1}))\frac{\partial x_t(\phi;\epsilon_t,x_{t+1})}{\partial\phi}\right.$$
$$\left.+ U_t^{\theta,\phi}(x_t(\phi;\epsilon_t,x_{t+1})) + \nabla_\phi r_{t+1}^{\theta,\phi}(x_t(\phi;\epsilon_t,x_{t+1}),x_{t+1})\right].$$
$$\square$$

Note that in the simplest case where we take $h_t$ as the filtering statistics and detach $h_t$ at each time step, these are reflected in the corresponding gradient computations and function updates. In this case, $\nabla_\phi\mathcal{L}_t(\theta,\phi) - \nabla_\phi\mathcal{L}_{t-1}(\theta,\phi)$ reduces to

$$\nabla_\phi\mathbb{E}_{q_t^\phi(x_t|y^t)q_t^\phi(x_{t-1}|y^{t-1},x_t)}\left[r_t^{\theta,\phi}(x_{t-1},x_t)\right]$$
$$+ \mathbb{E}_{\lambda(\epsilon_{t-1})\lambda(\epsilon_t)}\left[T_{t-1}^{\theta,\phi}\left(x_{t-1}(\phi;\epsilon_{t-1},x_t(\phi;\epsilon_t))\right)\frac{dx_{t-1}(\phi;\epsilon_{t-1},x_t(\phi;\epsilon_t))}{d\phi}\right]$$
$$+ \mathbb{E}_{q_t^\phi(x_t|y^t)q_t^\phi(x_{t-1}|y^{t-1},x_t)}\left[U_{t-1}^{\theta,\phi}\left(x_{t-1}\right)\right] - \mathbb{E}_{q_{t-1}^\phi(x_{t-1}|y^{t-1})}\left[U_{t-1}^{\theta,\phi}\left(x_{t-1}\right)\right]$$

since $q_{t-1}^\phi(x_{t-1}|y^{t-1})$ has been detached. Note that the terms on the last line represent the difference between two expectations of the $U_{t-1}^{\theta,\phi}(x_{t-1})$ function. We refer to this approach as "Ours (TU)". Without the final line, the objective reduces to the same objective as the non-amortized case, and we refer to this approach as "Ours (T)". We demonstrate the applicability of these methods on the Chaotic RNN task. For both the forward filtering and backward smoothing networks, we use a MLP with 2 hidden layers and 100 neurons in each layer. As shown in Table 3, the amortized networks are able to achieve similar filtering accuracy as the non-amortized case. Both of the proposed methods achieve lower errors during training time and test time.

Table 3: Root Mean Squared Error between filtering mean and true state on the CRNN task using amortized models. For test time errors, we rerun the trained models from the start of data without further optimization. The resulting errors are comparable to those of the non-amortized models.

| $d_x$ | | AELBO-1 | AELBO-2 | Ours (T) | Ours (TU) |
|---|---|---|---|---|---|
| 5 | Filter RMSE (train time) | 0.1158±0.0011 | 0.1039±0.0006 | **0.1032±0.0004** | **0.1031±0.0002** |
| | Filter RMSE (test time) | 0.1170±0.0022 | 0.1064±0.0012 | **0.1048±0.0005** | **0.1056±0.0013** |

### D.3 Semi-Amortized Approach

An alternative approach to amortization is to return to the exact same gradient computations as in the main paper. However, instead of $\phi_t$ corresponding directly to the statistics of $q_t^{\phi_t}$, it corresponds to the parameters of the RNN and MLPs which produce $q_t^{\phi_t}$ statistics from the observations $y^t$. For each time step, we run the inner optimization and 'overfit' the RNN/MLPs to the current set of observations. Overfitting in this context means that the networks are optimized to produce as accurate as possible $q_t^{\phi_t}$ statistics for this time step (just as in the main paper), but since only the current time step is considered in the optimization, they are not forced to generalize to other time steps and $y$-observations.

This may seem contradictory with the aims of amortization, namely using computational work spent during previous time steps to reduce the inference load at the current time step. However, if the RNN/MLPs are initialized at their optimized values from the previous time step, the previous optimization cycles can be thought of as a type of pre-training. This makes the inner optimization problem progressively easier in terms of computation required for a certain level of accuracy.

We demonstrate this idea using the linear Gaussian application where the distance to the true filtering distributions can be calculated analytically. We use an RNN and MLPs to generate $q_t^{\phi_t}$ statistics as described above and optimize each $\phi_t$ for 100 steps at each time step. Figure 6 plots the absolute error in the mean of $q_t^{\phi_t}(x_t)$ (averaged over dimension) versus time step for 5 different points within each inner optimization routine. Looking at the zero shot performance (at the start of each time step, before any optimization) we see that over time, the amortization networks are able to produce more and more accurate statistics without any updates using the current observations. This shows that this naive approach to amortization can indeed provide useful cost savings in the long-run.

Using this approach, we were able to reproduce the model learning results shown in Figure 1b but with fewer iterations per time step. This was also achieved for the Sequential VAE example using a convolutional RNN to encode video frames and MLPs to generate variational statistics. A visually plausible transition function could be successfully learned in this semi-amortized fashion.

### D.4 Gradient Computations

All approaches to amortization require computing gradients of the form $\frac{dx_t}{d\phi}$ with $x_t$ being sampled using the reparameterization trick with statistics from $q_t^{\phi}(x_t)$. When we use an RNN to calculate these statistics, calculating this derivative requires backpropagating through all previous observations $y^t$. This results in a linearly increasing computational cost in time. To avoid this, we detach the RNN state $h_{t-H}$ from the computational graph at some fixed window into the past, $H$. When we roll out the RNN to calculate statistics from time $t$, we simply initialize at $h_{t-H}$ and treat it as a constant. When the algorithm proceeds to the next time step, $h_{t+1-H}$ is then kept constant at its most recent value during the previous time step. More sophisticated methods for online training of RNNs are also possible, we refer to [31] for a survey.

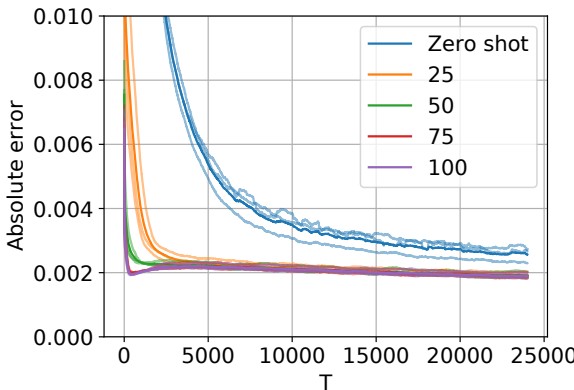

Figure 6: Absolute error of the mean of $q_t^{\phi_t}(x_t)$ averaged over dimension versus time step. 5 different points in each inner optimization process are plotted, zero shot is before any optimization steps, 25 is after 25 optimization steps, and so on. Different seeds are shown as translucent with the mean over seed shown in full color. The plateau in absolute error is due to the inherent limitations of stochastic gradient descent with a fixed learning rate. To contextualize the absolute error value, states and state transitions are on the order of $\sim 0.1$. The lines are smoothed using a uniform kernel of width 1000.

# E    Broader Impact

We propose a generic methodology for performing online variational inference and parameter estimation. Historically, filtering has been used in a huge variety of applications, some with large societal impacts. The same filtering algorithms can both help predict future weather patterns but could also be used in weapons guidance systems. Our current methodology remains in the research stage but as further developments are made that make it more practically applicable, it is important to fully consider these possible societal effects.