# OpenReview forum: "Online Variational Filtering and Parameter Learning"
_NeurIPS.cc/2021/Conference — NeurIPS 2021 Oral_

### Official Review · Reviewer_iuPB · 2021-07-12

**Rating:** 8
**Confidence:** 2

**Summary:**

This paper proposes a variational filtering scheme for state space models that enables recursion to successively estimate the ELBO to achieve efficient inference. Experiments are performed on linear Gaussian filtering, RNN estimation, and sequential VAEs.


**Limitations And Societal Impact:**

yes

**Main Review:**

tldr: I tend to weakly accept this paper, although I would strongly encourage the authors to make the connection to reinforcement learning way more explicit in the paper.

edit: after the author(s)' rebuttal, I tend to accept the paper. Thank you for the clarifications.

## Originality

*Are the tasks or methods new?* Yes, I haven't seen any sort of recursive filtering mechanism for state space models like this before.

*Is the work a novel combination of well-known techniques?* Yes, this is probably a novel combination of variational filtering with the variational interpretation of reinforcement learning models in the sense of [Levine, '18](https://arxiv.org/abs/1805.00909) and [Fellows et al, '18](https://arxiv.org/abs/1811.01132). Note that this is a positive comment.

*Is it clear how this work differs from previous contributions?* Yes, it's pretty clear how it differs from other approaches in the state-space models literature.

*Is related work adequately cited?* Yes, overall, I think so. See discussion + the papers listed above as well.

## Quality

*Is the submission technically sound?* Yes, it seems to be. The proofs seem to be mostly algebraic book-keeping (which is fine).
- I would have liked to have more details about the function approximators that use $\hat S_{t+1}$ and $\hat T_{t+1}$ in lines 161-180.

*Are claims well supported (e.g., by theoretical analysis or experimental results)?*
Yes, see comment above. The experimental results suggest that the method works pretty well.

*Are the methods used appropriate?*
Yes, overall. Not knowing the literature well I'd like to know how KRR is used in line 165.

*Is this a complete piece of work or work in progress?* This looks like a complete work.

*Are the authors careful and honest about evaluating both the strengths and weaknesses of their work?* Mostly, see discussion.

## Significance

*Are the results important?* Yes, better variational inference techniques for state space models are likely to encourage their usage in practice, while the technical connection to the Bellman recursion (I think) is probably important.

*Are others (researchers or practitioners) likely to use the ideas or build on them?* Yes, probably.

*Does the submission address a difficult task in a better way than previous work?* Yes.

*Does it advance the state of the art in a demonstrable way?* Yes.

*Does it provide unique data, unique conclusions about existing data, or a unique theoretical or experimental approach?* A unique theoretical / practical approach.

## Discussion

- While I agree that the connection to reinforcement learning is there (I think this is closely related to the Bellman recursion and dynamics models), the authors really need to do some more work to formalize this connection as otherwise, it's really lost on the reader and is a bit of a weakness.
    - RL can be constructed as probabilistic (and really variational) inference as in [Levine, '18](https://arxiv.org/abs/1805.00909) and succeeding works, so this work could help bridge the gap between these types of interpretations of RL and state space models generically.
    - However, none of the connections are made explicit so it's a bit tricky to tell right now what the impact is. Improving these connections would significantly improve the paper.
    - specifically, my concerns are about the paragraphs in lines 122 (I think you're referring to the bellman recursion but not sure),

### Questions

- While I understand that the time complexity is constant per iteration, it still seems like there is $t$ separate operations that much be computed. Does this mean that as the number of time steps increases, the method slows down over time?
    - Or is there only a single pass being computed through the sequence?

- In Fig. 1b why does the quality of the variational approximation get worse as the number of steps?
    - Is this overfitting or the quality of the variational approximation breaking down?

- How are the number of steps of SGD / Adam chosen in the inner loop?

- How are the summary models (either krr or mlp) initialized and with what sort of data? These are the models using $\hat S$ and $\hat T$?
    - Why doesn't computation for the KRR slow down over time?

- Is fig 1a just the first 10 time steps? Is there a tradeoff between "time per iteration" and "accuracy" here? If so, how is the tradeoff chosen?

### Writing Comments

20: network[s]

24: [estimating?] "online inference"

29: ref for "notoriously unreliable" as a non-expert in this field, i'd like to see a reference to verify.


**Time Spent Reviewing:**

6

---

> ### Author Response · Authors · 2021-08-10
> **Response to Reviewer iuPB (Part 1)**
>
> We thank the reviewer for their review and helpful suggestions.
>
> We will certainly make the link to reinforcement learning and dynamic programming more explicit in an update to the paper. The paragraph at L122 is indeed referring to the Bellman recursion. Our recursive equations can be thought of as Bellman recursions with suitably defined rewards, states and actions, but contrary to the standard approach, our recursions operate forward in time rather than backwards. We provide a technical exposition of all these links in a comment below (which we will also include in an update to the paper). We focus on our more general response to your review here.
>
> > _While I understand that the time complexity is constant per iteration, it still seems like there is $t$ separate operations that much be computed. Does this mean that as the number of time steps increases, the method slows down over time? Or is there only a single pass being computed through the sequence?_
>
> Indeed there is only a single pass through the sequence (we assume the method is applied online to one arbitrarily long time sequence) so the cost of each iteration does not increase as the number of time steps increases.
>
> > _In Fig. 1b why does the quality of the variational approximation get worse as the number of steps? Is this overfitting or the quality of the variational approximation breaking down?_
>
> We note that in Fig. 1b we are plotting the error in model parameters hence the slight increase in error at the end is not the variational approximation breaking down. Indeed the full joint variational distribution is increasing in dimension in time so inevitably the discrepancy with the true joint will increase. However, marginally for each filtering distribution we do not see an accumulation of error see e.g. Fig. 1a. Model parameter estimation is a complex problem and RMLE itself is not perfect even though it is analytic. Our approximations on top of RMLE add a small amount of error but it does not blow up and is comparatively better than the other baselines.
>
> > _How are the summary models (either krr or mlp) initialized and with what sort of data? These are the models using $\hat{S}$ and $\hat{T}$? Why doesn't computation for the KRR slow down over time?_
>
> The gradient summary models are initialized by regressing on a simulated dataset that only involves the first observation. For $\hat{S}_1$, this dataset is as on L162 with $\hat{S}_0$ defined to be 0 and for $\hat{T}_1$ the dataset is as on L167-168 with $\hat{T}_0$ again defined as 0 (see footnotes). The computation for KRR does not slow down because to perform the regression at each time step we just need to generate a fixed size simulated dataset (see Alg. 1). This dataset only depends on the current observation and so does not require looking back into the past (this effect is accounted for by including the previous regression model in the equations). There is then a fixed cost linked solely to the dataset size to complete the kernel regression, hence it does not increase with time (see also the response to all reviewers).
>
> > _How are the number of steps of SGD / Adam chosen in the inner loop?_
>
> > _Is fig 1a just the first 10 time steps? Is there a tradeoff between "time per iteration" and "accuracy" here? If so, how is the tradeoff chosen?_
>
> Regarding the number of steps of SGD/Adam in the inner loop, there is indeed a tradeoff between time per iteration and accuracy. If the time per iteration used to optimize the parameters $\phi_t$ at time $t$ is very small, then the accuracy won't be as high as when it is larger; the accuracy being limited in any case by the variational family used. In practice, we would recommend using as much time per iteration as the application allows. Further, we can usually observe convergence during preliminary runs allowing us to set the learning rate and number of iterations accordingly. In Fig. 1a it is just the first 10 steps, the tradeoff here heavily favoring accuracy. We should also note that we do not see an accumulation of error if we run for longer than 10 steps.
>
> > _29: ref for "notoriously unreliable" as a non-expert in this field, i'd like to see a reference to verify._
>
> We cited Kantas et al. (2015) in our paper which features discussion on the unreliability of these methods, we will add this additional reference to this section of the paper.
>
> > _Not knowing the literature well I'd like to know how KRR is used in line 165._
>
> In Li et al. (2020) KRR is used to estimate gradients of the log-likelihood for training a latent variable model. They exploit the conditional mean solution for MSE regression (see also A.1 in our appendix) avoiding the need for an encoding distribution. However, they do not make use of any recursive equations as we do here.
>
> > _I would have liked to have more details about the function approximators that use $\hat{S}\_{t+1}$ and $\hat{T}\_{t+1}$ in lines 161-180._
>
> We again thank the reviewer for this suggestion. For extra details, we refer to our response to all reviewers and Algorithm 1 in the paper. We will also include a more concrete example for the regression that is happening in the paper.
>
> **References**
>
> Kantas, N., Doucet, A., Singh, S. S., Maciejowski, J., and Chopin, N. (2015). On particle methods for parameter estimation in state-space models. Statistical Science.
>
> Li, W. K., Moskovitz, T., Kanagawa, H., and Sahani, M. (2020). Amortised learning by wake-sleep. ICML.

---

> > ### Author Response · Authors · 2021-08-10
> > **Response to Reviewer iuPB (Part 2)**
> >
> > We layout here the link between the recursions present in our method and RL. For comparison, we first state the standard recursion in RL. The total reward we would like to optimize is
> >
> > $$
> >     J(\phi) = \mathbb{E}\_{\tau \sim p\_\phi} \left[ \sum\_{t=1}^T r(s\_t, a\_t) \right]
> > $$
> >
> > This is taken with respect to the trajectory distribution which is dependent on the policy $\pi\_\phi$
> >
> > $$
> >     p_\phi(\tau) = P(s_1) \pi_\phi(a_1|s_1) \prod_{t=2}^T P(s_t|s_{t-1}, a_{t-1}) \pi_\phi(a_t|s_t)
> > $$
> > The value function is then defined as the expected sum of future rewards when starting in state $s_t$ under policy $\pi\_\phi$. It follows the Bellman recursion
> >
> > $$
> >     V\_t(s\_t) = \mathbb{E}\_{s\_{t+1}, a\_t \sim P(s\_{t+1}|s\_t, a\_t) \pi\_\phi(a\_t|s\_t)} \left[ r(s\_t, a\_t) + V\_{t+1}(s\_{t+1})\right]
> > $$
> >
> > $J(\phi)$ is then just the expected value of $V\_1$ taken with respect to the first state distribution.
> >
> > $$
> >     J(\phi) = \mathbb{E}\_{s\_1 \sim P(s\_1)} \left[ V\_1(s\_1)\right]
> > $$
> >
> > For our application, we would like to instead have a *forward* recursion. A natural forward recursion appears when we consider an *anti-causal* graphical model for RL, where $s\_t$ depends on $s\_{t+1}$ and $a\_{t+1}$; i.e. we consider the following *reverse*-time decomposition of the trajectory distribution
> >
> > $$
> >     p\_\phi(\tau) = P(s\_T) \pi\_\phi(a\_T|s\_T) \prod\_{t=T-1}^{1} P(s\_t|s\_{t+1}, a\_{t+1}) \pi\_\phi(a\_t|s\_t).
> > $$
> >
> > We define a new value function which is the sum of previous rewards
> >
> > $$
> >     V'\_{t+1}(s\_{t+1}) = \mathbb{E} \left[ \sum\_{k=1}^{t+1} r(s\_k, a\_k)\right]
> > $$
> >
> > It follows a corresponding forward 'Bellman' recursion
> >
> > $$
> >     V'\_{t+1}(s\_{t+1}) = \mathbb{E}\_{\pi\_\phi(a\_{t+1}|s\_{t+1}) P(s\_t| s\_{t+1}, a\_{t+1})} \left[ r(s\_{t+1}, a\_{t+1}) + V'\_t(s\_t)\right]
> > $$
> >
> > $J(\phi)$ is then the expected value of $V'\_T$ taken with respect to the final state distribution
> >
> > $$
> >     J(\phi) = \mathbb{E}\_{s\_T \sim P(s\_T)}\left[ V'\_T(s\_T)\right]
> > $$
> >
> > This forward 'Bellman' recursion is non-standard in the literature but is useful when we adapt it for our application. We define  $s\_t = x\_t$, $a\_t = x\_{t-1}$ and $P(s\_{t}|s\_{t+1}, a\_{t+1}) = \delta(s\_{t} = a\_{t+1})$. The 'reward' is defined as
> >
> > $$
> >     r(s\_t, a\_t) = r(x\_t, x\_{t-1}) =  \log \frac{f(x\_t|x\_{t-1}) g(y\_t|x\_t) q\_{t-1}^{\phi\_{t-1}}(x\_{t-1})}{q\_t^{\phi\_t}(x\_t) q\_t^{\phi\_t}(x\_{t-1}|x\_t)}
> > $$
> >
> > $$
> >     r(s\_1, a\_1) = r(x\_1, x\_0) =  \log \frac{p(x\_1) g(y\_1|x\_1)}{q\_1^{\phi\_1}(x\_1)}
> > $$
> >
> > Note $a\_1 = x\_0$ has no meaning here. The 'policy' is defined as the backward kernel
> >
> > $$
> >     \pi\_\phi(a\_t|s\_t) = q\_t^{\phi\_t}(x\_{t-1}|x\_t)
> > $$
> >
> > With these definitions, the trajectory distribution is
> >
> > $$
> >     p\_\phi(\tau) = q\_T^{\phi\_T}(x\_T) q\_T^{\phi\_T}(x\_{T-1}|x\_T) \prod\_{t=T-1}^1 q\_t^{\phi\_t}(x\_{t-1}|x\_t)
> > $$
> >
> > (Since $x\_0$ has no meaning in our application, the final $q\_1^{\phi\_1}(x\_0|x\_1)$ distribution that appears has no significance.)
> > With this formulation, the sum of rewards now corresponds to the ELBO which we would like to maximize with respect to $\phi$:
> >
> > $$
> >     \text{ELBO}\_T = \mathbb{E}\_{p\_\phi(\tau)} \left[ \sum\_{t=1}^T r(s\_t, a\_t)\right]
> > $$
> >
> > Just as in our anti-causal RL example, this can be broken down into a value function that summarizes previous rewards
> >
> > $$
> >     V'\_{t+1}(x\_{t+1}) = \mathbb{E}\left[ \sum\_{k=1}^{t+1} r(s\_k, a\_k)\right] = \mathbb{E}\_{q\_{t+1}^{\phi\_{1:t+1}}(x\_{1:t}|x\_{t+1})} \left[ \log \frac{p(x\_{1:t+1}, y^{t+1})}{q\_{t+1}^{\phi\_{1:t+1}}(x\_{1:t+1})}\right]
> > $$
> >
> > $$
> >     \text{ELBO}\_T = \mathbb{E}\_{q\_T^{\phi\_T}(x\_T)}\left[ V'\_T(x\_T)\right]
> > $$
> >
> > This follows a forward 'Bellman' recursion which appears as eq (6) in our paper
> > $$
> >     V'\_{t+1}(x\_{t+1}) = \mathbb{E}\_{q\_{t+1}^{\phi\_{t+1}}(x\_t|x\_{t+1})} \left[ \log \frac{f(x\_{t+1}|x\_t)g(y\_{t+1}|x\_{t+1})q\_t^{\phi\_t}(x\_t)}{q\_{t+1}^{\phi\_{t+1}}(x\_{t+1}) q\_{t+1}^{\phi\_{t+1}}(x\_t|x\_{t+1})} + V'\_t(x\_t)\right]
> > $$
> >
> > Since we would like to optimize the ELBO rather than just evaluate it, we don't make use of $V'\_{t+1}(x\_{t+1})$ directly. We instead differentiate the forward in time Bellman recursion to obtain our gradient recursions in the paper. To obtain eq (8) in the paper we differentiate with respect to $\theta$. To obtain eq (9) we differentiate with respect to $x\_{t+1}$, we then use $\frac{\partial}{\partial x\_{t+1}} V'\_{t+1}(x\_{t+1})$ to get an equation for $\nabla\_{\phi\_{t+2}} V'\_{t+2}(x\_{t+2})$.
> >
> > Our approach here is complementary to that of (Levine 2018, Fellows et al. 2019) but differs in the fact we focus on forward in time recursions allowing an online optimization of the ELBO. Levine (2018) and subsequent work focus on fitting RL into a probabilistic context whereas we take ideas from RL (recursive function estimation) to enable online inference. We note that Weber et al. (2015) also define suitable rewards to fit probabilistic inference into an RL framework but again only focus on backward Bellman recursions.
> >
> > **References**
> >
> > Levine, S. (2018). Reinforcement learning and control as probabilistic inference: Tutorial and review.
> >
> > Fellows, M., Mahajan, A., Rudner, T. G., \& Whiteson, S. (2019). VIREL: A variational inference framework for reinforcement learning. Advances in Neural Information Processing Systems
> >
> > Weber, T., Heess, N., Eslami, A., Schulman, J., Wingate, D., \& Silver, D. (2015). Reinforced variational inference. Advances in Neural Information Processing Systems Workshops.

---

> > > ### Comment · Reviewer_iuPB · 2021-08-11
> > > **Thanks**
> > >
> > > Thank you for the detailed clarifications. Although this is a bit outside of my domain of expertise, it's really nice to see the deep theoretical connection between the Bellman recursions and the ELBO in the state space model.

---

> > ### Comment · Reviewer_iuPB · 2021-08-11
> > **Thanks for these clarifications**
> >
> > Overall, these clarifications mostly answer my questions satisfactorily. Specifically, thank you for the pointer to Kantas et al, '15 as well as the detailed comments on KRR, the promise of the algorithm box, and the gradient summary models.
> >
> > I'm still a bit concerned about what does cause the breakdown at the end in Figure 1b, if it's not just the variational approximation breaking down. Your response raises another concern --- how does one ameliorate over-fitting in this online learning context?

---

> > > ### Author Response · Authors · 2021-08-23
> > > **Further Response**
> > >
> > > > _I'm still a bit concerned about what does cause the breakdown at the end in Figure 1b, if it's not just the variational approximation breaking down. Your response raises another concern --- how does one ameliorate over-fitting in this online learning context?_
> > >
> > > We have further investigated the experiment settings for Figure 1b to allay the reviewer's concerns. We performed a more extensive search over learning rate settings and doubled the running time to check for long term instabilities. We found that in all learning rate settings we tried, the method is stable and doesn't break down. For slower learning rate decays (such as those used in the paper) we face random walk-like behavior in the parameter estimates which is simply due to the nature of online recursive parameter estimation. The parameter update at each time step is based only on an estimate of $\nabla \log p(y_t|y^{t-1})$ and so natural variations in the observed $y_t$ value can cause parameters to sometimes move further from their true values. Even when analytically computing the updates (which we refer to as RMLE) there can still be long periods where the parameters are moving in the wrong direction. We rely on averaging in time to combat this and so when we run for a finite amount of time there can be cases where we are moving in the wrong direction just before we stop the simulation. In our new simulations, we found this behavior can be suppressed with a lower learning rate and faster learning rate decay. We can still reach similar levels of error by running the simulation for longer but now there is a near monotonic convergence in averaged parameter estimation error.
> > >
> > > This verifies that the behavior in Figure 1b is not due to the overfitting of the model parameters, variational distributions or function approximators but simply down to the learning rate schedule. We avoid overfitting the model parameters as we base our method on RMLE which has convergence guarantees to a stationary point of the average log likelihood. Like standard variational inference, maximizing the ELBO (or equivalently minimizing the KL divergence) also has an inherent maximum entropy regularization in the variational distributions. Finally, we mitigate overfitting for our function approximators by using L2 regularization (reported for KRR in the Appendix).

---

### Official Review · Reviewer_FJzo · 2021-07-16

**Rating:** 8
**Confidence:** 3

**Summary:**

This paper presents a variational inference algorithm for state-space models, that is suitable for online learning: at each time step, the variational posterior approximation can be updated to account for the new data in constant time. Parameters of the generative model can also be learned online. The algorithm is based on variational families $q_t^\phi$ that factor as $q^{\phi_t}(x_t \mid y_{1:t}) \prod_{i=t}^2 q^{\phi_{i}}(x_{i-1} \mid x_i, y_{1:i-1})$. The key point is that when a new datapoint $y_{t+1}$ is observed, the optimal backward kernels $p(x_{i-1} \mid x_{i})$ (which the $q^{\phi_i}(x_{i-1} \mid x_i)$ approximate) do not change for $i = 1,\dots,t$. The authors use this to justify freezing the parameters $\phi_i$ after time step $i$, so that at each time step, only the parameters $\phi_{t+1}$ must be learned. Besides this factorization, the other trick enabling online learning is a scheme for estimating gradients of the ELBO with respect to $\phi_{t+1}$ in constant time. This requires an additional approximation step, leading to biased gradient estimates. However, in experiments on several models and datasets, the authors show that their proposed technique can produce reasonable results in practice.


**Ethical Concerns:**

I don't see any ethical concerns -- although inference in state-space models can be applied to questionable ends (e.g. object/person tracking in surveillance videos), this is a methods paper, and the example applications presented here do not strike me as problematic.

**Limitations And Societal Impact:**

The authors include a limitations section, which I think is adequate, but could be expanded to discuss the issues raised in the main review. I did not see discussion of the potential negative societal impact, but the work is theoretical enough that I am not sure it is necessary to include.

**Main Review:**

This is a nicely written paper that presents (to my knowledge) a novel technique for online inference in state-space models, and validates it empirically in three compelling experiments. The exposition is clear and provides good intuition for the key insights behind Algorithm 1. I think the paper is of definite value and interest to the NeurIPS community and recommend acceptance.

However, I have a less clear picture of what exactly is happening theoretically. One question is what penalty  we should expect to incur (e.g., compared to batch variational inference) by using this online variant. Intuitively, we might expect that in the "online" Algorithm 1, the learned parameters $\phi$ would, assuming a sufficiently high number $M$ of gradient updates per time step, recover the optimal "batch" solution using the same variational family—i.e., the parameter settings you'd find if you optimized $L_t$ with respect to all of $\phi$ at once. But I believe this is not the case in general, because although $p_\theta(x_k \mid y^k, x_{k+1})$ does not depend on observations past time $k$, the optimal $q^{\phi}(x_k \mid y^k, x_{k+1})$ might. This is because, if $q^\phi(x_k \mid y^k, x_{k+1})$ cannot perfectly represent the true backward kernel, it will prioritize accurate backward transitions for the most probable values of $x_{k+1}$. Since the posterior on $x_{k+1}$ does change as new data comes in, the optimal $q$'s priorities also change. The experiments suggest that this may not be a problem in practice, perhaps because $q(x_k \mid x_{k+1})$ is sufficiently expressive to learn an accurate backward kernel for all $x_{k+1}$, or because in these particular datasets, the filtering and smoothing distributions over $x_{k+1}$ were sufficiently similar. But it might be nice to see some discussion of this issue, and/or an empirical validation that the online and batch variants learn similar variational approximations in practice.

A similar issue arises, I think, with the biased gradient estimates — L296-297 point out the importance of using sufficiently expressive function approximators to reduce the bias, but it also seems important that the training data used to optimize the approximators be from a good distribution. If I'm following correctly, the "training data" distribution for the approximators $\hat{T}$ and $\hat{S}$ are not equal to their "test time" distributions, which incorporate an additional data point. Could this introduce appreciable error in practice?

Overall, I suspect both these concerns are nitpicks -- but perhaps they could be signposted as assumptions the algorithm makes. To reiterate my overall positive impression of the work, I think the paper presents a clever idea with clear exposition and convincing experiments.

One very minor comment: in discussing [30], L222-223 state that "any" particle filtering technique will scale poorly when the state is high-dimensional, but this seems a harder claim to defend when the technique uses learned / well-adapted proposal distributions. (In any case, it is not a reason to dismiss PF in favor of VI; if you have a variational approximation to the posterior, you could use it within a particle filter as the proposal, and the accuracy should only improve.)

**Time Spent Reviewing:**

5 hours

---

> ### Author Response · Authors · 2021-08-10
> **Response to Reviewer FJzo**
>
> We thank the reviewer for their insightful review and deep engagement with the paper. We greatly appreciate the reviewer's acknowledgment of the paper's significance and their recommendation for acceptance. We here address the reviewer's concerns raised in the review.
>
> > _However, I have a less clear picture of what exactly is happening theoretically. One question is what penalty we should expect to incur (e.g., compared to batch variational inference) by using this online variant..._
>
> The comparison to the batched objective is a very good observation, we will include a discussion in the paper regarding this point. We agree that when considering all time steps $1:T$ jointly, this may indeed lead to different learned backward kernels. We have run some preliminary experiments and find that the learned joint variational distributions are indeed quite similar on the problem settings in the paper. Care must be taken for full batch training as there are some issues with stability for longer time sequences which our online objective circumvents. We will add these findings in the final version of this paper.
>
> In addition, our method is compatible with rolling back the variational distribution for a longer length $L$ (see e.g. L176-179) recovering an objective closer to the batched form. This offers a tradeoff between bias, variance and computation time. With a longer length, $L$, we would be able to optimize the backward kernel jointly at more times thus obtaining less bias, at the cost of larger variance and higher computation cost. When $L$ is equal to the length of the data, we effectively recover the batched objective without functional approximation.
>
> > _A similar issue arises, I think, with the biased gradient estimates -- L296-297 point out the importance of using sufficiently expressive function approximators to reduce the bias, but it also seems important that the training data used to optimize the approximators be from a good distribution. If I'm following correctly, the "training data" distribution for the approximators $\hat{T}$ and $\hat{S}$ are not equal to their "test time" distributions, which incorporate an additional data point. Could this introduce appreciable error in practice?_
>
> We also kindly appreciate the observation on distribution shift. This is an important point and we will add more discussion in the paper. We agree that the "training data" distribution $q^{\phi\_t}\_t(x\_t)$ for the approximators is not necessarily equal to their "test time" distributions $\int q^{\phi\_{t+1}}\_{t+1}(x\_t \mid x\_{t+1}) q^{\phi\_{t+1}}\_{t+1}(x\_{t+1}) dx\_{t+1}$. From a theoretical point of view, as discussed in L419-423 in the Appendix, given sufficiently expressive function approximators, the optimal solution of the supervised learning problem will be the exact function recursion we are looking for, for any "training data" distribution with enough coverage. Of course, in practice, the distribution will dictate the regions where the approximation is most accurate which is why we use $q^{\phi\_t}\_t(x\_t)$ as the default in our paper since it is the best available approximation of the state $x\_t$. In practice, we haven't found in our experiments that the distribution shift causes any training issues.
>
> > _One very minor comment: in discussing [30], L222-223 state that "any" particle filtering technique will scale poorly when the state is high-dimensional, but this seems a harder claim to defend when the technique uses learned / well-adapted proposal distributions. (In any case, it is not a reason to dismiss PF in favor of VI; if you have a variational approximation to the posterior, you could use it within a particle filter as the proposal, and the accuracy should only improve.)_
>
> We again thank the reviewer for this suggestion. Indeed basic particle filtering scales exponentially badly with the dimensionality, but it is possible to improve them by using well-learned proposal distributions. We will rewrite this part and will add that our approach could be combined with particle filtering and smoothing methods.

---

### Official Review · Reviewer_dXyo · 2021-07-16

**Rating:** 7
**Confidence:** 3

**Summary:**

The paper borrows the ideas from reinforcement learning to come up with an online inference and learning for state-space models. It focuses on the smoothing distribution instead of the usual filtering distribution for online learning. When performing variational EM with the new incoming data, the ELBO is recursively updated, and the gradients are estimated with a regressor. Experiments show good performance on various datasets.

**Limitations And Societal Impact:**

It would be great to state which requirement does not coincide with the usual online learning setup.

**Main Review:**

The paper proposes a novel online inference scheme for state-space models. The paper is generally clear and well-written. Here are some unaddressed points in my mind:
- It seems the (auxiliary) neural network for gradient estimation is learned in advance. If this is the case, in a non-stationary environment, will this estimation become outdated as the network is stationary?
To train the auxiliary neural networks for gradient estimation, an additional dataset is also needed. This seems not a desideratum of an online setup where an additional dataset is usually not available.
- Does it also work for model-based reinforcement learning since it borrows ideas from RL applications? If so, how does it perform?

**Time Spent Reviewing:**

2

---

> ### Author Response · Authors · 2021-08-10
> **Response to Reviewer dXyo**
>
> We would like to thank the reviewer for the review and the positive feedback. We very much appreciate that our method is considered novel and well described. We address below the main points from the review.
>
> > _It seems the (auxiliary) neural network for gradient estimation is learned in advance. If this is the case, in a non-stationary environment, will this estimation become outdated as the network is stationary? To train the auxiliary neural networks for gradient estimation, an additional dataset is also needed. This seems not a desideratum of an online setup where an additional dataset is usually not available._
>
> We would like to clarify that the additional dataset used to train the regressors  (i.e. $\hat{S}\_t$ and $\hat{T}\_t$) is *entirely simulated* and therefore no extra observations from the environment are required. This allows our method to be fully online. At each iteration, we train $\hat{S}\_t$ and $\hat{T}\_t$ using the simulated dataset that can be generated solely from the current time's variational distributions and also the previous gradient regressors $\hat{S}\_{t-1}$ and $\hat{T}\_{t-1}$. This means they can adapt to non-stationary environments and the estimation won't become outdated (see Algorithm 1 and also response to all Reviewers).
>
> > _Does it also work for model-based reinforcement learning since it borrows ideas from RL applications? If so, how does it perform?_
>
> Regarding the connection with model-based reinforcement learning, while we borrow value function ideas from RL, our paper focuses on variational inference and learning state evolution, instead of future reward estimation and action planning as in RL. We leverage the backward decomposition of the variational distribution to obtain a *forward* value function recursion (eq (6)), whereas, in standard RL, the forward decomposition is used to obtain a *backward* Bellman value function recursion (see also reply to Reviewer iuPB). This crucial difference allows our method to optimize the ELBO in a fully online manner, but may not transfer directly into an RL setting.
>
> On the other hand, we could potentially come up with a forward recursion for the value function in the RL setting by utilizing a time-reversed decomposition of the state-action Markov chain in a similar manner as in our paper (see reply to Reviewer iuPB for details). This may be useful for some model-based RL applications, and can serve as an interesting direction for future work.
>
> > _It would be great to state which requirement does not coincide with the usual online learning setup._
>
> We emphasize that our method has no limitations that prevent online learning, our method always has a constant cost per iteration. The additional dataset can be generated purely from the current observation already available and can be done in constant time. Furthermore, we use recursive function approximation to summarize previous time steps meaning we do not need to revisit the history, thus enabling our method to be fully online.

---

> > ### Comment · Reviewer_dXyo · 2021-08-16
> > **Thanks for the response**
> >
> > Thanks for clarifying the details! The work proposes a novel and interesting approach for online learning of state-space models, and I will increase my score to 7.

---

### Official Review · Reviewer_YSkX · 2021-07-16

**Rating:** 8
**Confidence:** 4

**Summary:**

This paper develops an algorithm for online variational inference and parameter learning in state-space models, where both model parameters and variational parameters are learned online with each new data point in the time series. The authors accomplish by approximating a backwards decomposition of the full posterior $p(x_{1:t}|y_{1:t}) = p(x_t|y_t) \prod_{k=1}^t p(x_k | y_{1:k}, x_{k+1})$ with a factored variational distribution $q_t(x_{1:t}) = q_t(x_t) \prod_{k=1}^{t-1} q_t(x_k | x_{k+1})$. This factored approximation is used to derive a (forward) recursive equation for computing the ELBO at time $t$, along with gradients of the ELBO with respect to the model and variational parameters. These recursive equations allow both the ELBO and its gradients to be estimated (using recursive learning of regressors) in an online manner, hence enabling online learning and inference. The authors evaluate this approach on two synthetic examples, and one example of video data, demonstrating that online inference successfully minimizes KL divergence to the true posterior as each new data point is observed, and that online learning successfully converges to plausible model parameters / transition dynamics, including in the video modeling task.

**Limitations And Societal Impact:**

The authors adequately addressed limitations and society impacts. With the extra space in the camera ready version, I would also suggest discussing their method in comparison with amortized variational inference in SSMs (e.g. using RNNs), which are also able to perform constant-time inference of new latent states after batch-wise training on a dataset of sequences. This means additional training cost, and potential overfitting, but potentially lower cost at inference time. It would be good to discuss the costs and benefits of each approach.

**Main Review:**

This was a nice paper to read. I thought it was clearly described and motivated, made use of the backwards decomposition of the posterior distribution in an insightful, original, and principled manner, and clearly demonstrated the utility of their approach relative to existing online baselines in a series of experiments. I just have a few clarificatory questions:

1. How fast is the regression step for $\hat T_t$ and $\hat S_t$? It was not immediately clear to me that this would take constant time, or exactly what kind of fitting was going on (it might be helpful to give a concrete example in the main text). I looked at the appendix, and it seems like it requires sampling some 500 and data points, and multiple iterations of fitting. This seems like it might be slightly too slow for some applications (e.g. real time robot tracking), and may require variable time if convergence of the regressor is desired.
2. Why is regression necessary at all, as opposed to Monte Carlo estimation of $T_t^{\theta,\phi}$ and $S_t^{\theta,\phi}$? Is it because they need to learned as functions of $x_t$ in order to be queried at different (sampled) values of $x_t$? Would it be possible to use (recursive) Monte Carlo approximations instead, avoiding the need for regression?
3. Are there potential issues with issues with the convergence of online learning, given that the recursive equations for $T_t^{\theta,\phi}$ and $S_t^{\theta,\phi}$ are not strictly followed once the parameters $\theta$ and $\phi$ are updated? e.g. $\hat S_{t-1}$ is learned using $\theta_{t-2}$, but $\hat S_t$ is learned using $\theta_{t-2}$. Perhaps there is RL theory addressing this?
4. How does this approach compare in speed / accuracy / computational cost versus amortized variational inference (e.g. learning an RNN as a recognition network)?

I also have the following minor comments:

- In the Abstract, I would personally avoid calling the decomposition / factorization "non-standard", because this gave me the impression that it might be unprincipled. "Backwards decomposition" would be more informative, and avoid that impression. Having read through the math fairly carefully, the proposed method is definitely principled in a way that I cannot say for many other papers that integrate variational inference with state space models.
- In Algorithm 1, there is a minor inconsistency between the update rules for $\phi_t$ and $\theta_t$ in terms of how the time-index subscripts are used.

Overall, I thought this was a solid paper. It could be made even stronger with a little more clarity on the regression steps, the inclusion of the AELBO baselines for the SVAE task, and additional comparisons of speed / accuracy to amortized inference methods, but the approach is adequately evaluated, well-motivated, and principled, and hence deserving of acceptance in my evaluation.

== Post-Rebuttal ==

Thank you to the authors for their detailed responses and clarifications. I believe that they have adequately responded to the questions raised by myself and other reviewers, and I will maintain my score of 8.

**Time Spent Reviewing:**

5

---

> ### Author Response · Authors · 2021-08-10
> **Response to Reviewer YSkX**
>
> We would like to thank the reviewer for their thorough evaluation of our work. We are especially pleased that the work is considered to be clearly described and well-motivated and that our use of the backward decomposition of the posterior distribution is regarded to be insightful, original, and principled. We address here the main points from the review.
>
> > _1. How fast is the regression step for $\hat{S}\_t$, $\hat{T}\_t$? ..._
>
> Thank you for the suggestion for a concrete example of the type of fitting we are carrying out, we will add this in an update to the paper. We would like to refer the reviewer to the response to all reviewers which gives more details regarding the regression step in Algorithm 1.
>
> Specific to this question, we note that when we use Kernel Ridge Regression, the constant computation cost of fitting is adequately low, and only depends on the size of the simulated training dataset (which is a hyperparameter of the method and can be adjusted based on computational budget). See Table 1 for a time comparison between the baselines.
>
> Further, if neural networks are used as the regression model, the network from the previous time step can be used as an initialization at the current time step which we found results in fewer iterations necessary for convergence as time goes on. The previous rounds of regression can be thought of as a type of pre-training due to the similarities in each regression problem thus resulting in this speed up.
>
> > _2. Why is regression necessary at all, as opposed to Monte Carlo estimation of $T\_t^{\theta, \phi}$ and $S\_t^{\theta, \phi}$?..._
>
> Indeed regression is necessary because we cannot know in advance which $x$ points the function will be queried at. Referring to Alg. 1, at time $t$ we evaluate $\hat{T}\_{t-1}$ and $\hat{S}\_{t-1}$ at $x_{t-1} \sim q_t^{\phi_t}(x_t)q_t^{\phi_t}(x_{t-1}|x_t)$. This distribution over $x_{t-1}$ is not available at time $t-1$ because it is a smoothing distribution using information from observation $y_t$. It is a good point that recursive Monte Carlo approximations could be used instead though this would require rolling all the way back to $t=1$ to be unbiased. As we assume arbitrarily long sequences, this makes the method no longer online and would have high variance. We can interpolate between these two extremes through rolling back part way and using a function approximation at a lag; see L176-179 for a discussion.
>
> > _3. Are there potential issues with issues with the convergence of online learning..._
>
> The use of old parameter values when training the function estimators is certainly something we must consider. We first note that this is not an issue if we assume the model parameters $\\theta$ are known and we only train $\\phi_{1:T}$. This is because it is possible for the function approximator, $\hat{T}\_t$, to perfectly approximate the true value, $T\_{t}^{\\theta, \\phi_{1:t}}$, given enough simulated training data since the fact we have a different $\phi_t$ at each time step is already considered in the updates.
>
> If we also learn the model parameters, then it is true that $\hat{S}\_t$ cannot match the true $S\_t^{\theta, \phi_{1:t}}$ value as it would require revisiting all previous states with the new updated $\theta$ value.
> For this reason, we borrow from Recursive Maximum Likelihood Estimation (RMLE) and update the current parameters using a difference of log likelihood gradients each computed with a filter that uses parameters $\theta_t$ at time $t$. There are proofs of convergence for RMLE based on a particle filter (see e.g. Tadić 2010, Tadić & Doucet 2020).
>
> > _4. How does this approach compare in speed / accuracy / computational cost versus amortized variational inference..._
>
> This is a very interesting question and we will update the paper to address it. There are two types of amortization we could consider here.
>
> We could learn an amortization network for the full sequence through batch-wise training on a dataset of similar sequences. This is likely to be more computational costly at training time especially for longer sequences, because of the need to backpropagate through increasingly long inference models, but can be more efficient at test time. However, this batched dataset of sequences may be difficult to obtain in scenarios where online learning is desired. Also, if the dataset is too small, this may result in potential overfitting whereas this is not an issue for our method which continually adapts to all incoming data. An alternative is to learn an amortized transition network in an online manner, and our work is an essential first step towards being able to achieve this form of amortization.
>
>
> Thank you for also pointing out the wording in the abstract and time-index issues, we will update these in the paper.
>
> **References**
>
> Tadić, V. B. (2010). Analyticity, convergence, and convergence rate of recursive maximum-likelihood estimation in hidden Markov models. IEEE Transactions on Information Theory.
>
> Tadić, V. B., & Doucet, A. (2020). Asymptotic properties of recursive particle maximum likelihood estimation. IEEE Transactions on Information Theory.

---

> > ### Comment · Reviewer_YSkX · 2021-08-22
> > **Thank you for these clarifications**
> >
> > Thank you for these helpful clarifications! My questions have been addressed, and I will be maintaining my score.

---

### Author Response · Authors · 2021-08-10
**Response to all Reviewers**

We would like to thank all the reviewers for their detailed analysis of our paper and very helpful reviews. We were pleased that all reviewers considered our work to be novel, well-motivated, clearly described and valuable to the NeurIPS community.

We would like to take the opportunity here to clarify one question regarding Algorithm 1 which has been raised by several reviewers.

During the regression step at time step $t+1$, we are simply performing supervised learning on a *simulated* dataset that we have generated afresh at the current time step. The inputs correspond to sampled $x\_{t+1}$ points from the most recent approximate filtering distribution $q\_{t+1}^{\phi\_{t+1}}(x\_{t+1})$. The outputs correspond to gradients evaluated at joint samples of $x\_{t+1}$, $x\_t \sim q\_{t+1}^{\phi\_{t+1}}(x\_t|x\_{t+1})$, along with the previous regression models at time $t$ evaluated at those $x\_t$ samples (see L162 and L167-168). We emphasize that this does not require any additional observations from the environment or any other secondary dataset in advance, it is purely generated from the data we already have available in this online setup.

Note that for both Kernel Ridge Regression (KRR) and neural networks, the computation time for the regression (including dataset generation) is constant in time. For KRR, this simply involves creating a kernel matrix from the training data input points. For neural networks, they are trained for a certain number of training iterations, taking minibatches from the simulated dataset.

We address below the major comments made by each reviewer in individual responses. Other comments will also be addressed in the final version of the paper.

---

### Decision · Program_Chairs · 2021-09-27

**Decision:**

Accept (Oral)

**Comment:**

The authors propose a general scheme for approximation in state space systems where the variation distribution is conditioned _backwards_ in time.  This leads to an ELBO (and gradients) that can be computed online, without having to adjust previous terms.

The reviewers unanimously praised the clarity of the writing, and I was impressed that each reviewer was left with a very clear picture of the paper's contribution and relevance - a sign that the idea has been presented with great clarity.

One clarification that arouse during the discussion was around a regression that occurs at time step $t+1$, and whether the input/output pairs here are drawn in such a way that could introduce error. I'm happy with the authors' clarification, which I anticipate in the camera ready manuscript.

One in-depth discussion between a reviewer and the authors covered connections to reinforcement learning: the authors connect Bellman recursion to the ELBO updates. This discussion could make for a neat appendix to the paper.